## PROCEEDINGS A

climatology, oceanography, hydrology

sea level, climate change, ocean warming, land ice loss

**Author for correspondence:**
Anny Cazenave
e-mail: anny.cazenave@legos.obs-mip.fr

An invited Review to mark the election of Anny Cazenave to the fellowship of the Royal Society in 2021.

# Contemporary sea-level changes from global to local scales: a review

Anny Cazenave[1] and Lorena Moreira[2]

[1]Laboratoire d'Etudes en Geophysique et Oceanographie Spatiales, Toulouse, France
[2]International Space Sciences Institute, Bern, Switzerland

AC, 0000-0002-2289-1858

Sea-level variations spread over a very broad spectrum of spatial and temporal scales as a result of complex processes occurring in the Earth System in response to natural variability of the climate system, as well as to external forcing due to natural phenomena and anthropogenic factors. Here, we address contemporary sea-level changes, focusing on the satellite altimetry era (since the early 1990s), for which various observing systems from space and *in situ* allow precise monitoring of sea-level variations from global to local scales, as well as improved understanding of the components responsible for the observed variations. This overview presents the most recent results on observed global and regional sea-level changes and on associated causes, focusing on the interannual to decadal time scale. Recent progress in measuring sea level at the coast are presented. Finally, a summary of the most recent sea-level projections from the Intergovernmental Panel on Climate Change is also provided.

## 1. Introduction

The sea-level climate variable is of major interest for a broad range of applications, depending on spatial and temporal scales. At global spatial scale and interannual to decadal time scale, the global mean sea level (GMSL) is recognized as a leading indicator of global climate change because it reflects changes occurring in different compartments of the climate system. Present-day sea-level rise and its acceleration [1,–3], currently estimated

by high-precision satellite altimetry, are mostly driven by anthropogenic global warming, i.e. more specifically ocean warming-induced thermal expansion, and ice mass loss from glaciers, Greenland and Antarctica [4]. As recommended by the Global Climate Observing System (https://gcos.wmo.int), the World Meteorological Organization (WMO) now includes the GMSL as one of the seven key global indicators of present-day climate change [5]. Regular assessments of the GMSL budget (i.e. comparison of observed GMSL change with sum of components) have been recently performed (e.g. by the WCRP Global Sea Level Budget Group [2], the IPCC/Intergovernmental Panel on Climate Change Special Report on the Oceans and the Cryosphere [4] and the IPCC 6th Assessment Report/AR6 [6]; and references therein), showing that for the last three decades (i.e. the satellite altimetry era), the GMSL budget is closed within data uncertainties. This indicates that sea level and the contributions can now be precisely quantified, with no significant gap in knowledge or major instrumental problems in the observing systems used to estimate the GMSL and its components. Contemporary GMSL evolution and sea-level budget are now routinely included in the WMO annual report on the status of the global climate [5], in the annual climate and ocean reports of the Copernicus services of the European Union (www.copernicus.eu), as well as in the state of the climate report yearly published by the National Ocean and Atmosphere Administration (NOAA, USA) [7].

At regional scale, satellite altimetry has revealed that sea-level rise can significantly deviate from the global mean. Accurate monitoring of regional sea-level changes and understanding the underlying causes (mostly non-uniform ocean warming and salinity changes) are of primary importance for detection/attribution studies (i.e. discriminate between anthropogenic forcing and natural/internal climate variability) and detection of time of emergence of the forced (anthropogenic) signal depending on regions.

At local (coastal) scale, sea level relative to the land surface is the quantity of practical interest for understanding the societal impacts of sea-level change. It is now well recognized that coastal sea-level change can be substantially different from open ocean sea-level rise because near the coast, small-scale processes superimpose on the global mean and regional sea-level components [8]. In addition, in many coastal zones, vertical land motions caused by ground subsidence amplify the climate-related sea-level rise [9]. Relative coastal sea-level rise is a major driver of shoreline retreat and erosion that acts in combination with other processes of natural (e.g. storm surges and cyclones) and anthropogenic (sand extraction, urbanization, land use) origin [10].

A large number of articles have been published on sea level and components in recent years, including in the successive IPCC assessments, thus making a synthesis of this abundant literature a difficult exercise. However, in this overview article, we have tried to summarize the most recent results about sea-level changes and causes from global to local scales, focusing on the high-precision altimetry era (since 1993) and on the interannual to decadal time scale. After the introductory section, §2 presents the GMSL observations and causes of rise and acceleration. In §3, the regional sea-level changes and contributions are discussed. In §4, the driving factors causing observed global and regional changes and the current view about the respective roles of anthropogenic forcing versus natural and internal climate variability are addressed. Coastal sea-level changes are discussed in §5, while most recent projections of future sea-level changes are summarized in §6. Conclusions and perspectives are provided in §7.

## 2. GMSL: a synthesis of recent research

### (a) Observations

Since the early twentieth century, two tools are used to measure sea level: tide gauges and satellite altimeters. Tide gauges measure relative sea level with respect to the Earth crust on which they are installed, hence cannot distinguish between vertical land movements (VLMs) and height variations of the water column. Satellite altimeters measure absolute sea level in a geocentric reference frame in which is referred the satellite orbit. Although the spatial distribution of tide

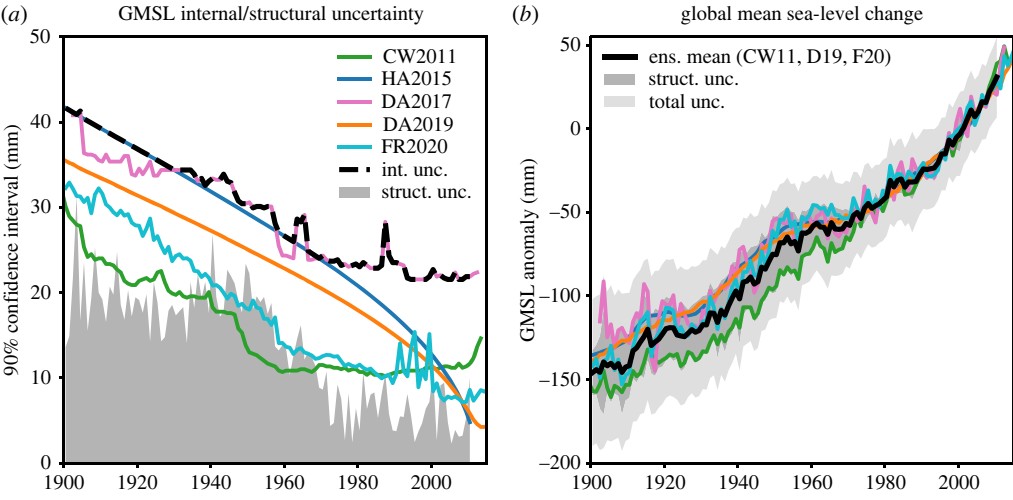

**Figure 1.** Mean sea-level reconstructions from 1900 (VLMs corrected). (*a*) The internal uncertainties of individual sea-level time series from different authors (coloured lines; see below) and of the ensemble mean (dashed black). The grey shaded region represents the ensemble structural uncertainty due to the chosen approach for reconstructing sea level from individual time series. (*b*) The individual sea-level reconstructions and the ensemble mean (green: [15]; pink and orange: [16,17]; blue: [18]; turquoise: [19]; black: ensemble mean) (figure from [14]). (Online version in colour.)

gauges is limited to continental coastlines and islands, and the coverage drastically decreases back in time, tide gauge-based sea-level records provide the invaluable historical reference to which present-day sea-level rise can be compared. Numerous studies have attempted to reconstruct a VLM-corrected mean sea-level time series since 1900 using coastal tide gauges records combined with additional statistical information. Two broad categories of methods have been developed: (1) a 'virtual' station approach based on regional influence area to minimize the weight of regions densely covered by tide gauges (e.g. the northeastern coast of North America and western Europe) and (2) the use of dominant spatial patterns deduced from satellite altimetry for the recent decades, to account for the open ocean regional variability not sampled by tide gauges. A third category of past sea-level reconstructions combines tide gauge data with synthetic sea-level data deduced from historical runs of coupled climate models (prior to 2005) (e.g. [11,12]). The different methodological approaches are reviewed in Oppenheimer *et al.* [13] and Palmer *et al.* [14]. Together with the uncertain VLM correction applied to the tide gauge data, the different methods lead to substantial dispersion in the reconstructed past mean sea-level curves, especially for the early decades of the twentieth century. The recent study by Palmer *et al.* [14] used five recent reconstructions in an ensemble approach to quantify the twentieth century mean sea-level rise and its uncertainty. They estimate the mean sea-level elevation of $12 \pm 5$ cm between 1901 and 1990 with a mean rate of $1.3 \pm 0.6$ mm yr$^{-1}$ over the period. This historical sea-level rate estimate agrees well with the one provided by Oppenheimer *et al.* [13], of $1.4 \pm 0.6$ mm yr$^{-1}$ over the same time span (figure 1).

Since the early 1990s, sea level is measured by high-precision satellite altimeters. In concept, radar altimetry is among the simplest of remote sensing techniques. Two basic geometric measurements are involved: (1) the distance between the satellite and the sea surface (called 'range') is determined from the round-trip travel time of microwave pulses emitted downward by the satellite's onboard radar and reflected back from the ocean surface and (2) the three-dimensional position of the satellite in space in a geocentric reference frame using different tracking systems such as satellite laser ranging, Doppler Orbitography Radiopositioning Integrated by Satellite (DORIS) and Global Navigation Satellite System (GNSS). Combining the range measurement and the satellite altitude above a fixed reference (typically a mathematically

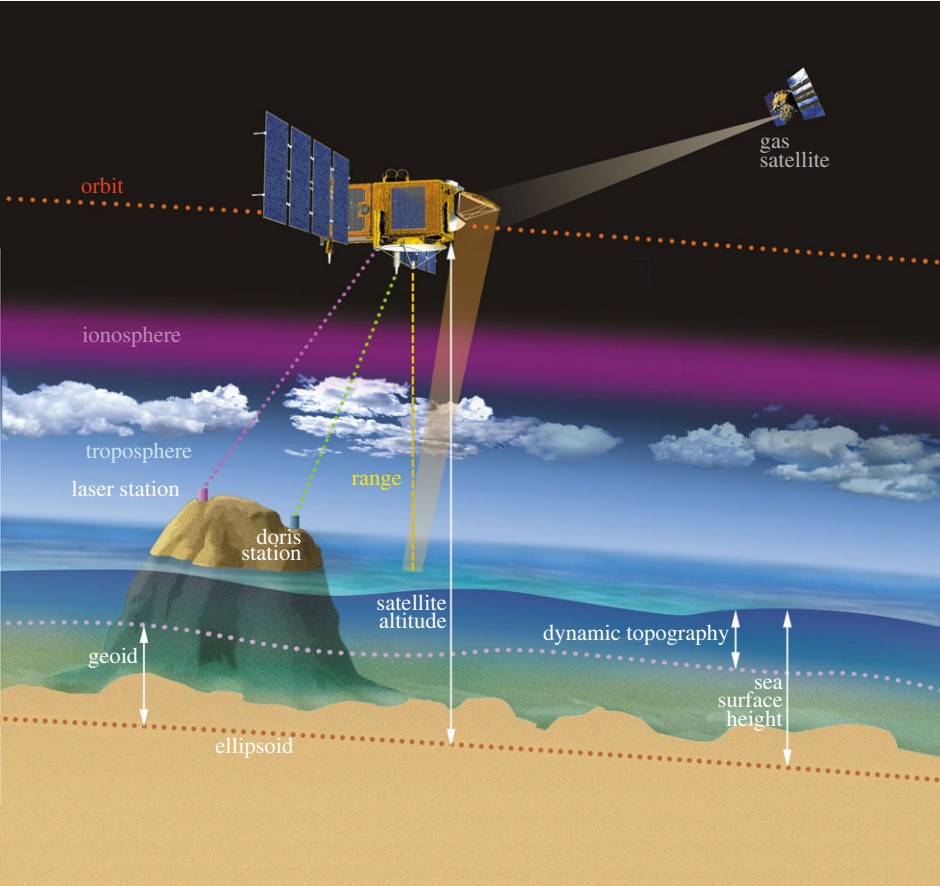

**Figure 2.** Satellite altimetry-based measurement of the sea surface height. The geoid is an equipotential surface of the Earth gravity field that coincides with the mean sea level at rest. The dynamic topography is the sea surface height above the geoid. Other quantities are defined in the text (source AVISO, https://www.aviso.altimetry.fr). (Online version in colour.)

defined ellipsoid that represents the mean shape of the Earth) allows mapping the sea surface topography along the satellite track with respect to the reference ellipsoid. Figure 2 shows a schematic representation of classical nadir altimetry. The estimated sea surface height needs to be corrected for various factors due to atmospheric delay, instrumental drifts and bias between successive altimetry missions. Other corrections due to geophysical effects, such as solid Earth, pole and ocean tides are also applied (an overview of the satellite altimetry technique can be found in [20]).

The first high-precision altimetry mission, Topex-Poseidon, was launched in August 1992. It has been followed by a series of missions with identical orbital characteristics (chosen for optimizing the sea surface height measurements): Jason-1, 2, 3 and Sentinel-6 Michael Freilich missions launched in 2001, 2008, 2016 and 2020, respectively. These represent the so-called reference missions for sea-level studies due to their high accuracy and long-term stability. They cover the 66° N–66° S domain and are complemented by other missions in different orbits at higher latitudes (up to 98.5° N/S) (e.g. ERS-1/2, GFO, Envisat, CryoSat-2, SARAL/AltiKa and Sentinel 3-A/B).

Different groups routinely process the measurements of these altimetry missions and provide either a GMSL record or gridded sea-level time series (or both), e.g. the University of Colorado, https://sealevel.colorado.edu; NOAA, http://www.star.nesdis.noaa.gov/sod/lsa/

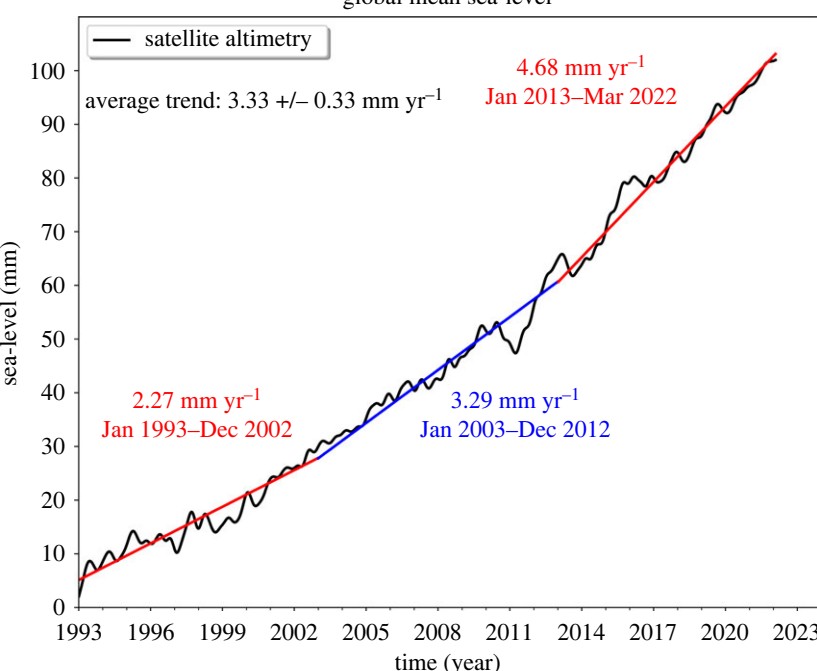

**Figure 3.** Global mean sea-level time series (black curve) from multimission altimetry (data from AVISO, https://www.aviso. altimetry.fr) over January 1993 to March 2022. The coloured straight lines represent linear sea-level trends over three successive time spans.

SeaLevelRise/LSA_SLR_timeseries_global.php; GSFC/Goddard Space Flight Center, http://podaac-ftp.jpl.nasa.gov/dataset/MERGED_TP_J1_OSTM_OST_GMSL_ASCII_V2 in the USA. In Europe, sea-level products essentially come from AVISO (https://www.aviso.altimetry.fr) and the marine and climate change services of the Copernicus programme of the European Union (https://www.copernicus.eu). The Copernicus marine service (https://www.marine. copernicus.eu) uses all available altimetry missions in orbit at any epoch in order to increase the spatial and temporal resolution of the gridded data. This product is well suited for mesoscale ocean circulation applications. The dataset delivered by the Copernicus Climate Change service (C3S, https://www.climate.copernicus.eu) provides gridded sea-level time series based at any time on a reference mission plus a complementary mission to increase the spatial coverage. In this product, sea-level trends are derived from the reference missions only because of their superior long-term stability. Thus the C3S product is well suited for climate-relevant sea-level studies (https://cds.climate.copernicus.eu/cdsapp#!/dataset/satellite-sea-level-global?tab=\penalty-\@Moverview).

Figure 3 shows the global mean sea-level curve over January 1993 to March 2022. It is based on the most up-to-date reprocessing of altimetry missions and includes a correction for the instrumental drift occurring during the first 6 years of the Topex/Poseidon mission [21]; see also [2] as well as a glacial isostatic adjustment (GIA) correction of $-0.3\,\mathrm{mm\,yr^{-1}}$ [22,23].

The rate of global mean sea-level rise over this 29-year time span is $3.33 \pm 0.33\,\mathrm{mm\,yr^{-1}}$. The $0.3\,\mathrm{mm\,yr^{-1}}$ uncertainty (90% confidence level) is based on Ablain *et al.* [24], who quantify the sources of errors affecting all components of the altimetry system (drifts, biases and noises), hence the determination of the GMSL. In figure 3, a clear acceleration is also observed. The rate of rise has increased from $2.3\,\mathrm{mm\,yr^{-1}}$ over 1993–2002 to $4.7\,\mathrm{mm\,yr^{-1}}$ over 2013–2022, hence by a factor of 2 (see §2.3 for the GMSL acceleration estimates).

## (b) Causes of global mean sea-level rise

In this section, we summarize the most up-to-date knowledge about the various contributions causing the global mean sea-level rise over the altimetry era. The regional variability will be discussed in §3.

In terms of global mean, there are three main contributions to the global mean sea-level rise: ocean thermal expansion, land ice loss (glacier melting and ice sheet mass loss) and terrestrial water storage changes. Quantification of these three contributions has evolved through time and it is only recently that they can be directly estimated using different observing systems. Prior to the altimetry era, their estimation was based on modelling only or on the combination of models and observations (e.g. [12,25]).

### (i) Contributions to the GMSL

GMSL changes essentially result from steric sea level and ocean mass changes. Steric sea-level change results from change in ocean density (assuming that local ocean mass does not change) [26]. Steric sea level is made of two parts: (1) the thermosteric component due to change in sea water temperature T and (2) the halosteric component due to change in water salinity S. Both thermosteric and halosteric components are estimated by vertically integrating, from a given water depth to the sea surface, T and S anomalies due to local changes in temperature or salinity in the sea water column with respect to a reference state. The thermosteric component is also called thermal expansion. Steric sea level is the sum of the thermosteric (temperature change only) and halosteric (salinity change only) contributions respectively called thermosteric and halosteric sea level. It is worth noting that the global mean halosteric contribution is zero because the total salt content of the ocean is constant [26,27]. Thus, in principle, only thermal expansion needs to be considered when estimating the steric contribution to the GMSL (if the data coverage is really global). Fresh water addition to the ocean (e.g. from land ice melt) only changes the global mean ocean mass [26].

The global mean ocean mass component, also called barystatic sea level [26], includes the effects of glaciers melting, Greenland and Antarctica ice sheets mass loss, and terrestrial water storage changes.

*Steric sea level.* Until the mid-2000s, the majority of ocean temperature data have been retrieved from shipboard measurements. These include vertical temperature profiles along research cruises supplemented by measurements from ships of opportunity. These *in situ* temperature measurements were mostly based on expandable bathy thermographs. They are limited to the upper 700 m ocean depth, but some profiles down to the ocean bottom have also been collected (e.g. [28]). Although the coverage improved through time, some regions remained under-sampled, in particular, the southern hemisphere oceans and the Arctic area [29].

The implementation of thousands of autonomous floats of the international Argo programme during the first half of the 2000s drastically improved the coverage [29–31]. More than 80% of the initially planned full deployment of the Argo float programme was achieved during the year 2005, with quasi-global coverage, except for a few regions (Arctic Ocean, Indonesian Seas). At present, about 4000 floats provide systematic temperature and salinity data down to 2000 m depth, over the 60° S–60° N latitude range (figure 4).

Different gridded time series of temperature/T and salinity/S data down to 700 m (prior to Argo) and 2000 m (Argo era), as well as thermal expansion time series based on vertical integration of the T/S data, have been produced by different groups for the altimetry era. Details on these databases can be found in recent articles (e.g. among others, [–2,32,34]). To give an order of magnitude, the WCRP Sea Level Budget Group [2] reports a rate of thermal expansion of $1.3 \pm 0.4 \, \mathrm{mm \, yr^{-1}}$ over 1993–2015 based on the ensemble mean of 11 different datasets. An update by Camargo *et al.* [34], based on an ensemble mean of 10 datasets, indicates a value of $1.36 \pm 0.1 \, \mathrm{mm \, yr^{-1}}$ over 1993–2017 for the contribution of the 0–2000 m ocean depth layer (figure 5). The deep ocean contribution (below 2000 m) is assumed to be small so far, of the order

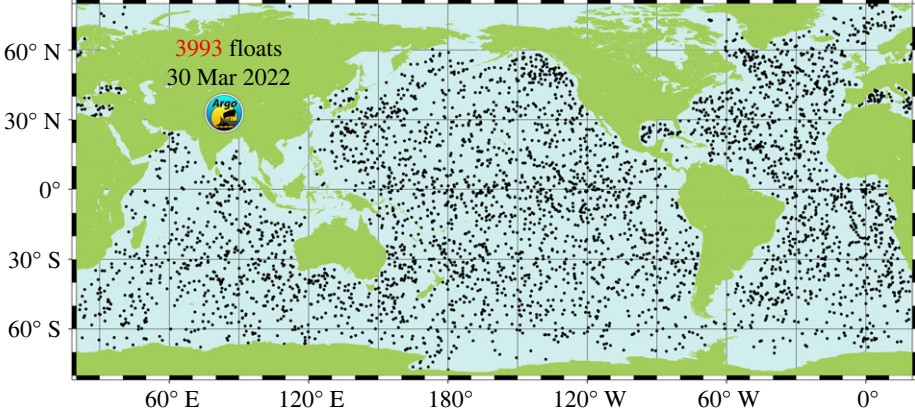

**Figure 4.** Current Argo floats coverage (source https://argo.ucsd.edu/about/status/). (Online version in colour.)

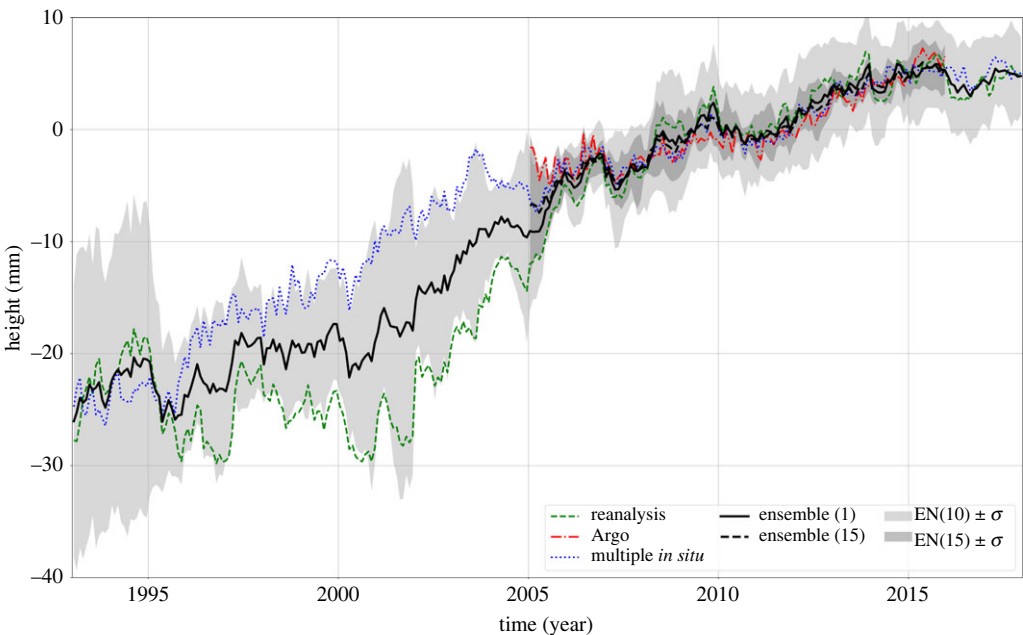

**Figure 5.** Steric sea-level evolution for 1993–2017 based on different datasets. The black curve represents the ensemble mean of the considered datasets, and the grey shaded area the 1-sigma standard deviation (figure from [34]). (Online version in colour.)

of 0.1 mm yr$^{-1}$ [28]. Due to continuing global warming, the ocean heat content has continued to increase over the past few decades, especially since the early 2000s [33,35,36]. This is also the case of the steric sea level that follows a roughly similar behaviour as the ocean heat content [32].

*Glaciers.* Global glacier mass changes are derived from *in situ* mass and length measurements as well as from remote sensing. Four types of space-based measurements are used: (1) glacier elevation changes by differencing between two epochs digital elevation models (DEMs) based on satellite imagery, (2) surface flow velocities from Synthetic Aperture Radar Interferometry (InSAR) for determination of mass fluxes, (3) laser and radar altimetry for measuring elevation changes and (4) direct mass change estimates by space gravimetry (see [37] for an overview). Modelling of glacier mass balance driven by climate observations has also been developed,

especially for estimating the glacier contribution during the historical period [38,39]. Most recent estimates of global glacier mass balance show clear evidence of total mass loss (e.g. [40–42]). The study by Hugonnet *et al*. [43] gives a total glacier mass loss of $267 \pm 16 \, \text{Gt yr}^{-1}$ (gigatonnes per year) over 2000–2019 on the basis of the DEM's approach. This corresponds to $0.74 \pm 0.04 \, \text{mm yr}^{-1}$ sea-level rise (SLR). A value of $281.5 \pm 30 \, \text{Gt yr}^{-1}$ ($0.75 \pm 0.08 \, \text{mm yr}^{-1}$ SLR) is reported by Ciraci *et al*. [44] on the basis of mass change estimates from the GRACE and GRACE Follow-On space gravimetry missions over 2002–2019 [45]. These studies also indicate significant acceleration of global glacier mass loss over the last two decades.

*Ice sheets.* To estimate the mass balance of the ice sheets, three main methods are used: (1) measurement of elevation changes of the ice surface over time either from satellite imagery or altimetry (e.g. [46–49]); (2) the mass budget or Input–Output Method (IOM), which involves estimating the difference between surface mass balance and ice discharge from InSAR (e.g. [50,51]); and (3) direct mass change measurements by GRACE-GRACE Follow-On space gravimetry (e.g. [52,53]). Comparisons between the three methods are provided by the IMBIE Team [54,55] (see also [56–60]). Similar to glaciers, all studies report ice sheet mass loss over the altimetry era, with significant acceleration during the last decade. For example, Mouginot *et al*. [51] indicate an average Greenland mass loss of $286 \pm 20 \, \text{Gt yr}^{-1}$ ($0.79 \pm 0.05 \, \text{mm yr}^{-1}$ SLR) for 2010–2018 compared with $187 \pm 17 \, \text{Gt yr}^{-1}$ ($0.51 \pm 0.05 \, \text{mm yr}^{-1}$ SLR) for the 2000s. Similarly, Rignot *et al*. [50] show an increased mass loss for Antarctica, of $252 \pm 26 \, \text{Gt yr}^{-1}$ ($0.69 \pm 0.07 \, \text{mm yr}^{-1}$ SLR) over 2009–2017, whereas the 1999–2009 average is estimated to be $166 \pm 18 \, \text{Gt yr}^{-1}$ ($0.45 \pm 0.05 \, \text{mm yr}^{-1}$ SLR).

It is now well established that the recent ice mass loss from the ice sheets partly results from accelerated glacier flow along some coastal margins of the ice sheets and further iceberg discharge into the surrounding ocean [4,6]. Dynamical instabilities triggered by ocean warming in regions where coastal glaciers are grounded below sea level (this is especially the case in West Antarctica) cause thinning and subsequent break-up of floating ice tongues or ice shelves that buttressed the glaciers. This results in rapid grounding line retreat and accelerated glacier flow (e.g. [61–63]).

In West Antarctica (the largest contributing region to current Antarctica mass loss), the Thwaites glacier (one of the most unstable glaciers, flowing into the Amundsen Sea Embayment) is of particular concern as it is retreating at an alarming rate since the early 1990s because of warming of the surrounding ocean that erodes the buttressing ice shelves and directly melts the grounded ice [64]. Since the early 2000s, the Thwaites Glacier has had a net loss of more than 1000 billion tons of ice. Its contribution to global mean sea-level rise is around 4%. Recent research shows that if Thwaites Glacier fragmentation and retreat continues, leading to total collapse, it has the potential to increase the contribution to the global mean sea-level rise by up to 65 cm [65–68].

Dynamical coastal instabilities explain almost all ice mass loss for Antarctica, while for Greenland ice mass loss equally results from coastal dynamical instabilities and decreasing surface mass balance [4].

*Terrestrial waters.* Terrestrial water storage change results from climate variability and direct human interventions (construction of dams on rivers and ground water pumping for crop irrigation). At global scale, total (vertically integrated) land water storage change is either directly measured by GRACE space gravimetry [69–71], or estimated by global hydrological models (e.g. [72–75]). Reconstructions combining GRACE data with meteorological data have also been performed for estimating the terrestrial water contribution prior to GRACE [76]. Direct human interventions on the hydrological cycle are also estimated using available information on dam building and ground water extraction combined with modelling (e.g. [77–79]).

Studies dedicated to quantify the terrestrial water contribution to the GMSL display an important dispersion in the estimates. The WCRP Global Sea Level Budget Group [2] discussed this issue and concluded that even the sign of this component was unknown. For example, based on GRACE data analysis over the continents, Reager *et al*. [70] found a negative contribution to the GMSL of $-0.33 \pm 0.16 \, \text{mm yr}^{-1}$ over 2002–2014. This is in contrast to model-based estimates and mass budget approaches that indicate a positive contribution to sea level (e.g. [80–83]), of

the same order of magnitude but opposite sign compared with GRACE results. This is mostly due to the strong interannual variability of the land water storage time series that prevents from accurately estimating the (small) linear trend over relatively short time spans because it is completely hidden by the interannual signal. Moreover, as discussed in Scanlon *et al.* [73], lack of direct measurements (except over the GRACE time span) and important discrepancies between global hydrological models are additional limitations to accurately estimating this component. Nevertheless, Caceres *et al.* [75] recently revisited this question and showed that most up-to-date hydrological modelling accounting for human activities and validated against GRACE data indeed lead to a positive contribution of terrestrial waters to the GMSL, of $0.41 \pm 0.11 \, \mathrm{mm \, yr^{-1}}$ over 2003–2016.

### (ii) Global mean sea-level budget over the altimetry era

Assessing the GMSL budget over the altimetry era, i.e. comparing the altimetry-based GMSL time series with the sum of components, has been the object of many publications during recent years (e.g. [2,3,80,83,87]). Assessing the GMSL budget closure is important for detecting temporal changes in the GMSL or in its components [3,83,87], or missing contributions (e.g. deep ocean warming [81,82,88]), for process understanding, validating climate models and detecting systematic bias or drifts in the observing systems (e.g. altimetry, Argo, GRACE and GRACE-FO) involved in the sea-level budget [89,90]). Recently published studies have shown that until the end of 2016, the GMSL budget is closed within respective uncertainties of the sea-level budget components (e.g. [4,80]). The steric and land ice components have increased over the altimetry era. Horwath *et al.* [80] show quasi-closure of the sea-level budget over January 1993–December 2016 based on specific dataset computations in the framework of the Climate Change Initiative project of the European Space Agency (ESA) (https://climate.esa.int/en/projects/sea-level-budget-closure/). In the latter study, thermal expansion and barystatic (ocean mass) component amount to $1.15 \pm 0.12 \, \mathrm{mm \, yr^{-1}}$ (38% of the GMSL trend estimated to $3.05 \pm 0.24 \, \mathrm{mm \, yr^{-1}}$) and $1.75 \pm 0.12 \, \mathrm{mm \, yr^{-1}}$ (57%), respectively. The mass component can be decomposed into $0.64 \pm 0.03 \, \mathrm{mm \, yr^{-1}}$ (21% of the GMSL trend) for glaciers, $0.60 \pm 0.04 \, \mathrm{mm \, yr^{-1}}$ (20%) for Greenland, $0.19 \pm 0.04 \, \mathrm{mm \, yr^{-1}}$ (6%) for Antarctica and $0.32 \pm 0.10 \, \mathrm{mm \, yr^{-1}}$ (10%) for terrestrial waters. Figure 6 shows for January 1993 to December 2016, the evolution of individual contributions, their sum and the altimetry-based GMSL (data from [80]).

Overall, over the altimetry era, ocean thermal expansion and glaciers have been the two largest contributions to the GMSL rise, immediately followed by the Greenland contribution. If one considers glaciers, Greenland and Antarctica all together, the land ice contribution dominates. The numbers gathered in the SROCC [4] and IPCC AR6 [6] give slightly different percentages because they are based on different published results and different time spans. For example, the IPCC AR6 reports 46%, 44% and 10% for the thermal expansion, total land ice and terrestrial water contributions, respectively, over 1993–2018. For comparison, the SROCC gives 49%, 47% and 4% for the same components over 1993–2015. All studies stress that terrestrial water storage change remains the least well-known contribution to the GMSL rate. As explained above, this mostly results from the dominant interannual variability of the time series, as well as from lack of direct measurements (except over the GRACE time span) and important discrepancies between global hydrological models [73,75].

The sea-level budget is more accurate over the 2005–present time span because of the use of GRACE space gravimetry to directly estimate the ocean mass increase rather than the individual mass components (e.g. [91]), and also because of the use of Argo, with quasi-global coverage and measurements down to 2000 m. For the period 2005–2016, Horwath *et al.* [80] estimate the thermal expansion and the ocean mass contributions to 34% and 66%, respectively, with the total land ice contributing by approximately 55%.

For the most recent years, closure of the sea-level budget has been questioned [89,90]. The recent period indeed coincides with instrumental problems encountered by GRACE at its very life end (in 2017) and by its successor GRACE Follow-On launched in 2018 [92]. Hence inaccurate

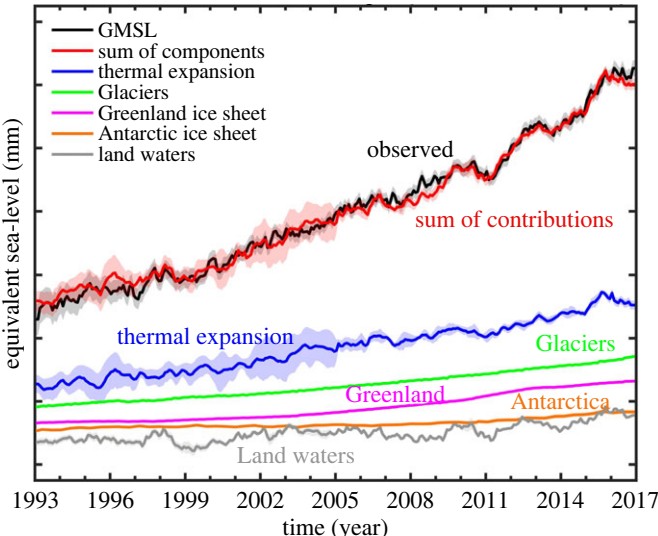

**Figure 6.** Global mean sea-level budget over 1993–2016. The bottom curves refer to individual components. The black and red curves represent the altimetry-based GMSL time series and the sum of contributions, respectively. Shaded areas represent 1-sigma errors  (data from [80]). (Online version in colour.)

GRACE data have been suspected to be responsible for non-closure of the sea-level budget since 2017. However, most recent investigations have pointed out the role of wrong Argo salinity data used to estimate the steric component [90] and a slight drift of the radiometer onboard the Jason-3 altimeter satellite (launched in 2016), from which the wet troposheric correction applied to altimeter data is estimated [93]. Besides, validation of recent GRACE and GRACE Follow On data have been shown to be free from any significant drift [49,89]. These recent investigations conclude that the GMSL budget can still be considered as closed within data uncertainties [93].

## (c) Acceleration of the GMSL

As shown in figure 3, the GMSL is accelerating [3,83,87,94,98]. Over the altimetry era, the estimated acceleration ranges from $0.084 \pm 0.025\,\mathrm{mm\,yr^{-2}}$ [3,87] for 1993–2017 (after correcting for the Pinatubo volcanic eruption) to $0.093 \pm 0.01\,\mathrm{mm\,yr^{-2}}$ [97] for 1991–2019 and $0.11 \pm 0.01\,\mathrm{mm\,yr^{-2}}$ [98] for 1993–2019. According to Ablain *et al*. [24], the uncertainty of GMSL acceleration estimates may not be better than $0.07\,\mathrm{mm\,yr^{-2}}$ (90% confidence level) for the 1993–2017 time span, a value about twice the dispersion range of the above reported values.

   Previous studies based on tide gauge data had already reported an acceleration in the mean sea-level rise, initiated in the late 1960s [17], a possible consequence of increased ocean heat uptake. For the last two decades, accelerated ice mass loss from Greenland and Antarctica mostly contribute to the observed GMSL acceleration [50,51,53], but as mentioned earlier, glacier mass loss also contributes [43]. The contributions to the acceleration of each component of the GMSL budget computed using data from Horwath *et al*. [80] over the time interval from 1993 to 2016 are the following: steric 19.9%; glaciers 19.3%; Greenland ice sheet 38.6%; Antarctica ice sheet 18.4%. The terrestrial water storage essentially contributes to the interannual variability.

## (d) Interannual variability of the GMSL

It is now well recognized that at interannual time scales, GMSL anomalies are mostly driven by internal climate variability, in particular by the El Niño-Southern Oscillation (ENSO) [99–101] and

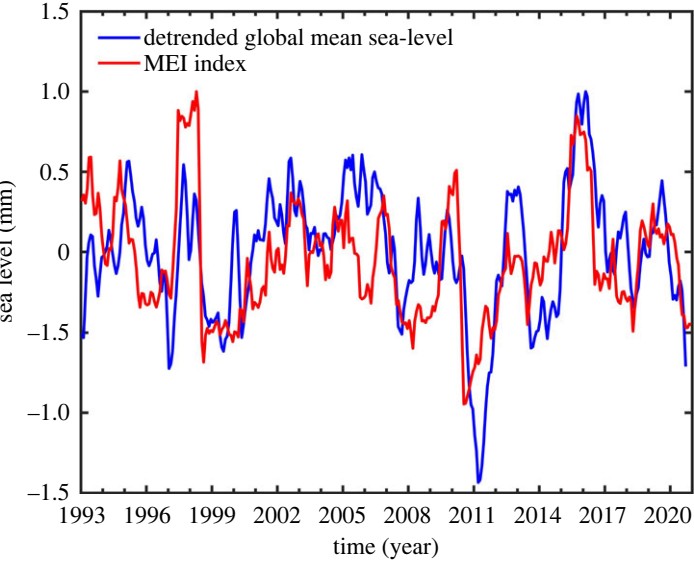

**Figure 7.** Interannual variability of the (quadratically detrended) global mean sea level (blue curve) with the Multivariate ENSO Index (MEI, red curve) superimposed. The global mean sea-level temporary increases in 1997–1998 and 2015–2016 are correlated with positive MEI anomalies and the occurrence of El Niño events. The opposite is observed during La Niña (e.g. in 2011). (Online version in colour.)

the Pacific Decadal Oscillation (PDO) [102,103] climate modes. Figure 7 displays the quadratically detrended GMSL over the altimetry era, on which is superimposed the Multivariate ENSO Index (MEI, a proxy of ENSO). Both time series are highly correlated. Further studies have shown that the temporary positive or negative anomalies encountered by the GMSL during ENSO events are caused by water mass exchange between river basins on land and tropical oceans [103–108]. During an El Niño, reduced rainfall over tropical river basins corresponds to increased precipitation over the tropical Pacific, hence temporary sea-level rise [101,105] and the opposite during La Niña (e.g. [106]). However, Piecuch & Quinn [109] showed that the steric component also contributes to the positive/negative GMSL anomalies during ENSO events. This was recently confirmed by Hamlington *et al*. [110,111], who demonstrated that both the steric and mass components equally contribute to the interannual variability of the GMSL, especially during El Niño and La Niña episodes (see also [112,113]).

# 3. Regional sea level

## (a) Observations

The global coverage of altimetry missions has allowed estimates of the regional rates of sea-level change. These are shown in figure 8*a* for the period January 1993 to August 2021 (last available data at the time of writing). We note that most regions display positive sea-level changes over the period. In some areas, the rate of rise exceeds the 3.3 mm yr$^{-1}$ global average. This is better seen when the global mean trend of 3.3 mm yr$^{-1}$ is removed at each mesh of the gridded dataset, as illustrated in figure 8*b* that shows regional sea-level trends with respect to the global mean trend.

## (b) Current process understanding of regional sea-level trends

On interannual to decadal time scales, sea-level change over a given oceanic region results from the global mean sea-level rise due to ocean warming, land ice melt and water exchange

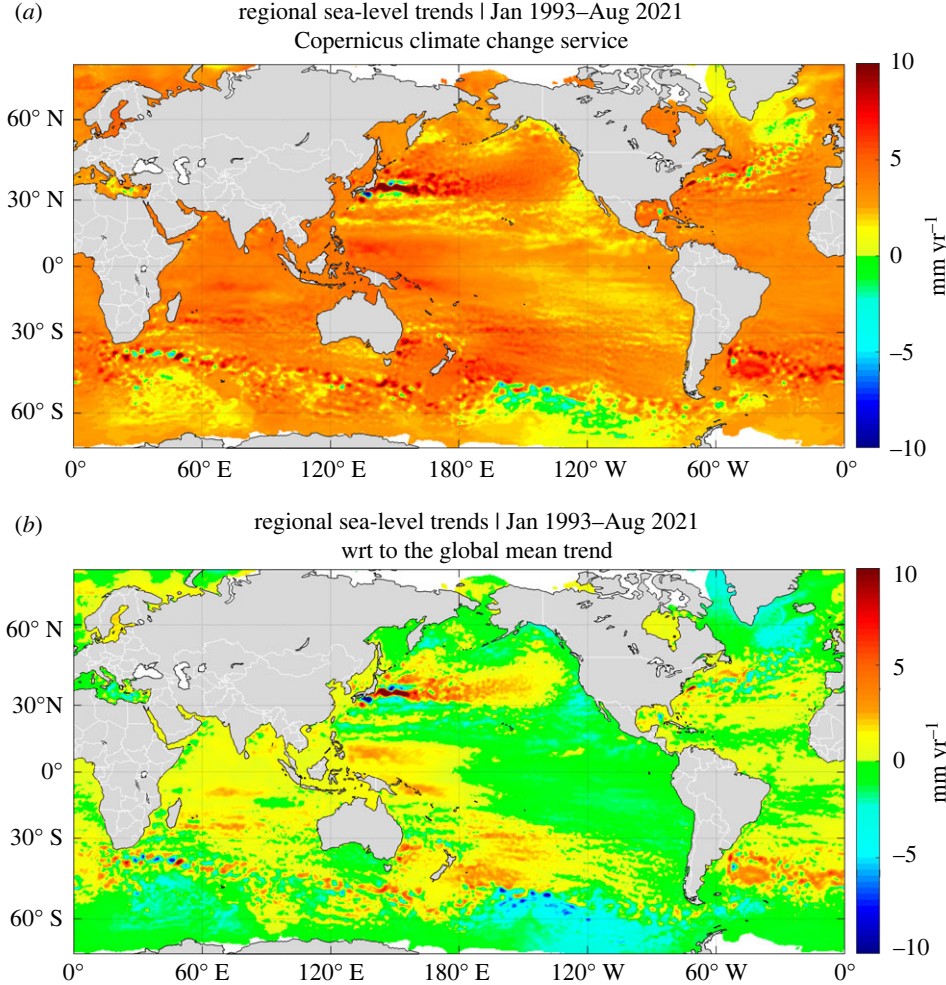

**Figure 8.** (*a*) Regional sea-level trends from January 1993 to August 2021, based on multi mission satellite altimetry with global mean trend included. (*b*) The same as (*a*) but with the global mean trend of 3.3 mm yr$^{-1}$ removed. In figure 8*b*, yellow/red and green/blue colours respectively correspond to regions where sea-level trends are larger or smaller than the global mean trend (data from the Copernicus Climate Change Service, https://www.climate.copernicus.eu). (Online version in colour.)

with continents, on which are superimposed geographical trend patterns caused by different processes: local/regional changes in sea water density due to changes in temperature and salinity (steric effects), redistribution of ocean water mass by the ocean circulation (called manometric component, [26]), atmospheric loading, and solid Earth's deformations and gravitational changes in response to mass redistributions caused by past and present-day land ice melt and land water storage changes. The latter factors have two components: the one associated with last deglaciation, i.e. GIA, and the other related to present-day land ice melt, and to a lesser extent to land water storage change (called GRD fingerprints [26]).

### (i) Regional steric changes

Regional steric changes result from regional changes in temperature (thermosteric component) and salinity (halosteric component).

The main forcing mechanisms causing regional steric changes are: (1) mechanical forcing from surface winds and (2) buoyancy heat and fresh water fluxes associated with variations in the

overlying atmospheric state [114,115]. Wind forcing causes variations in the ocean circulation that further redistributes heat and water masses. It has been shown that wind-driven steric changes dominate steric changes in most regions (i.e. [116]). This is particularly the case in the tropics where the interannual to decadal variations in steric sea level are essentially wind-driven (e.g. [117–119]). But wind forcing also plays a role in the extra tropics and at high latitudes [115]. While buoyancy forcing has a limited impact at low-latitude sea level, surface air-sea fluxes of heat and fresh water (due to surface warming and cooling of the ocean, and exchange of fresh water with the atmosphere and land through evaporation, precipitation and runoff) can be important in some oceanic areas, e.g. in the North Atlantic (Gulf Stream and North Atlantic subpolar gyre) [115].

Compared with steric changes, barotropic redistribution of mass plays a smaller role on interannual to decadal time scales, but not necessarily negligible (e.g. [11,120], in particular, at high latitudes and over shallow continental shelves (e.g. [121]). Over the altimetry era, regional sea-level patterns are dominated by steric changes (figure 9). In most regions, the thermosteric component by far dominates the halosteric one, except in high latitude areas, e.g. in the northeast Pacific, and particularly in the Arctic (e.g. [123,124]). In the coming decades, with expected increased land ice melt, the GRD fingerprints will also significantly contribute to regional sea-level trends.

On interannual to multidecadal time scales, the spatial trend patterns in (thermo) steric sea level are still largely driven by basin-scale natural (internal) climate modes (ENSO, PDO, Atlantic Multidecadal Oscillation (AMO), North Atlantic Oscillation (NAO), Indian Ocean Dipole (IOD), etc.) [114,125]. Wind stress changes on such time scales are directly related to the climate modes [125]. For example, as shown in several studies (e.g. [126]), sea level in the tropical Pacific oscillates from west to east with ENSO (with high/low sea level in the eastern/western part during El Niño and inversely during La Niña), in response to wind-forced baroclinic propagating waves. In the North Atlantic, surface wind and heat flux partly drive interannual to decadal sea-level fluctuations and are associated with the NAO (but changes in the Atlantic Meridional Ocean Circulation (AMOC) also contribute, [125]). In the tropical Indian Ocean, interannual to decadal variability in sea level is strongly influenced by ENSO and the IOD [125,127].

Figure 9 compares the regional sea-level trends observed by satellite altimetry over 2005–2020 with the Argo-based steric trends over the same time span (global mean trends removed). Similarity between the spatial patterns of the two maps clearly confirm that for the recent years, the steric component dominates the observed regional sea-level trends. Figure 9c also shows the difference in trends between the two maps. Significant signal still remains in this residual map. It combines the effect of data errors (both from altimetry and Argo) and additional physical processes (e.g. the fingerprints of present-day land ice melt; see §3b(iv)).

### (ii) Regional ocean mass changes

Regional ocean mass change (manometric component) essentially results from regional/local redistribution of water mass by the ocean circulation (see §3b(i)). Water mass addition to the ocean due to land ice melt and exchange of water with the continents (barystatic component) leads to rapid uniform mass change over the oceanic domain via a barotropic global adjustment occurring on short time scales (a few weeks) [128].

### (iii) Atmospheric loading

From seasonal to longer time scales, sea level reacts isostatically to first order to changes in atmospheric pressure changes [129]. The sea level responds as an inverted-barometer to atmospheric loading, i.e. the sea surface height increases (decreases) by 1 cm if the local surface pressure decreases (increases) by 1 mbar. The atmospheric loading component is quite small compared with the thermosteric one but non-negligible at high latitudes (e.g. in the Arctic where it can reach $0.3 \, \mathrm{mm \, yr^{-1}}$ on interannual to decadal time scales [130]). Atmospheric loading can be estimated using surface pressure data from atmospheric reanalyses. Note that

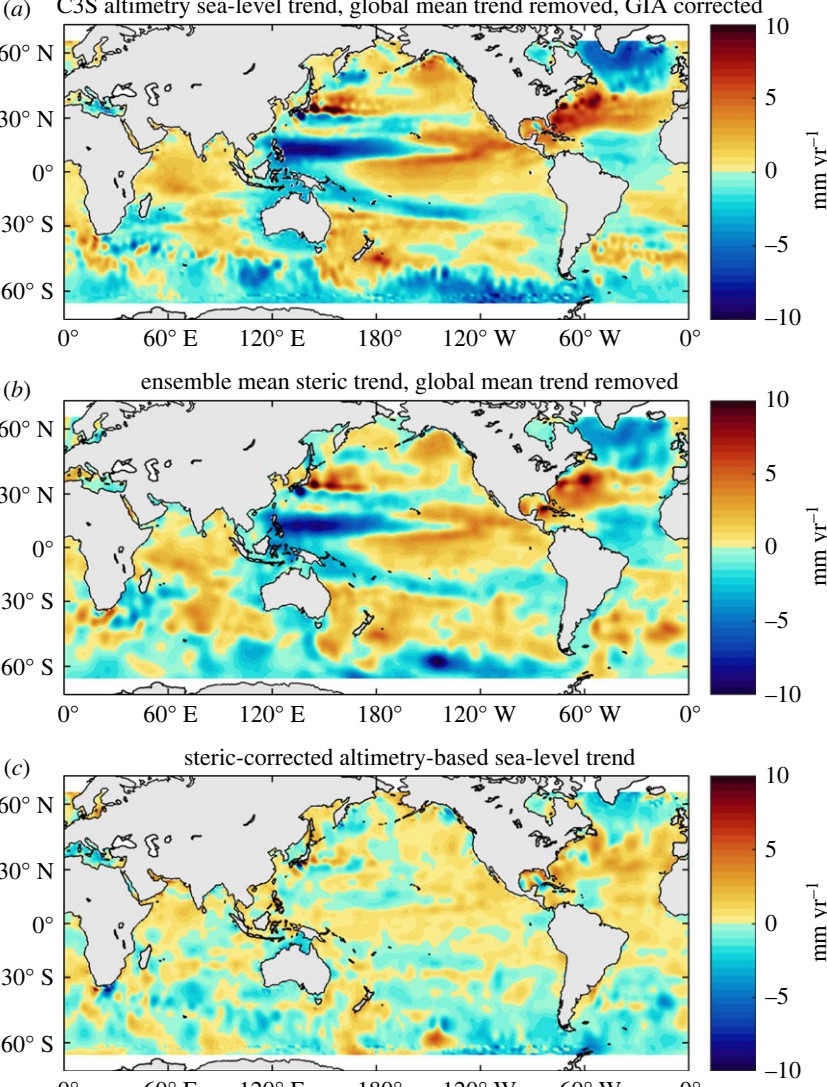

**Figure 9.** Regional altimetry trends over 2005–2020 (*a*; data from C3S, https://www.climate.copernicus.eu) and steric trends over the same time span (*b*). Global mean trends have been removed from both maps. The steric data are based on an ensemble mean of four different solutions (details in [122]). (*c*) The residual trend map (altimetry minus steric trends) (adapted from [122]). (Online version in colour.)

when geographically averaged over the oceanic domain the atmospheric loading effect vanishes because of mass conservation (i.e. it has no effect on the GMSL).

### (iv) Fingerprints of past and present-day land ice melt

Regional sea-level changes may also arise from the response of the solid Earth to past and present-day water mass exchange between continents and oceans. The GIA effect depends on Earth's mantle viscosity and deglaciation history while the response of the solid Earth to ongoing land ice melt essentially depends on the elasticity of the lithosphere and mantle, as well as amount and location of ice mass loss. These mass redistributions induce changes in Earth Gravity, Earth Rotation and cause elastic solid Earth Deformation [114,131,132]. These phenomena—so far

mostly predicted by modelling—give rise to complex regional patterns in sea-level change: sea-level drop in the immediate vicinity of the melting bodies but sea-level rise in the far field (e.g. along the northeast coast of America).

*Glacial isostatic adjustment.* As shown in different studies dedicated to solve the so-called sea-level equation (e.g. [22,23,133,135]), GIA—the visco-elastic response of the solid Earth and associated rotational change to last glaciation—is a long-term process that still impacts contemporary sea-level observations. In altimetry data (i.e. absolute sea level), GIA is a small signal in terms of global average (approx. $-0.3\,\mathrm{mm\,yr^{-1}}$ [22,23,136]). Its regional signature is mostly uniform, except in formerly glaciated high-latitude regions. Although small, GIA needs to be taken into account in altimetry-based global and regional sea-level studies.

*Sea-level fingerprints of present-day water mass redistribution.* Water mass redistribution from present-day melting of the ice sheets and glaciers, or from the hydrological cycle over the continents, causes changes in regional sea-level patterns, known as GRD sea-level fingerprints [26]. These sea-level fingerprints result from the changing gravitational attraction between ice and water bodies on land and the resulting mass change of the oceans, plus the associated deformations and rotational changes of the solid Earth, due to the changing load [23]. Ice sheet and glacier mass loss by far dominates the fingerprint patterns, causing sea level to fall near the ice bodies, while in the far field, sea level rises in order to conserve mass [23]. Several studies have theoretically computed the associated changes in (relative and absolute) sea level as well as in vertical land motions, solving the sea-level equation either assuming *a priori* ice sheet mass loss (e.g. [23,131,137]), or using realistic ice mass loss based on observations from the GRACE space gravimetry mission [138]. The GRD sea-level fingerprints have small amplitude (a few $\mathrm{mm\,yr^{-1}}$) compared with observed regional sea-level trends and steric trends. A few studies tried to detect the GRD fingerprints by looking at either ocean basin scale averages or coastal regions [139,140]. Moreira *et al.* [122] used altimetry-based sea-level maps corrected for steric effect for detecting the regional GRD fingerprint signature due to present-day water mass redistribution. They concluded that about 25% of the variability observed in the steric-corrected altimetry-based sea level in the vicinity of the Greenland ice sheet may be explained by the absolute sea-level fingerprints. But in other oceanic regions, the low signal-to-noise ratio of the observations prevents any clear detection, due to remaining errors in the steric data [141], and in the altimetry data [142].

## (c) Regional sea-level budget over the altimetry era

A few recent studies have assessed the closure of the basin-scale sea-level budget over the altimetry era (e.g. [12,143,145]; see also [110] for a review). Closure of the regional budget is only observed in some regions but not everywhere. For example, using altimetry, GRACE and Argo data over 2005–2015, Royston *et al.* [146] came to the conclusion that the regional budget cannot be closed in the Indian-south Pacific region.

To highlight the closure/non-closure of the regional sea level, figure 10 shows the spatial patterns of the first three modes of an empirical orthogonal function (EOF) decomposition of the altimetry-based sea level corrected for steric effects over 2005–2020 (same time span as in figure 9). The principal components are also displayed in figure 10 along with the time series of two climate modes: MEI and PDO. Mode 1 displays a dominant trend signal, positive in the tropical oceans, mid-North Atlantic and Northeast Pacific, and negative south of Greenland and in the Austral Ocean between South America and Antarctica. We cannot exclude that part of the residuals mode 1 reflects the signature of the GRD fingerprints. Mode 2 and 3 spatial patterns are significant in the central and eastern Pacific and Indian oceans (and also in the Austral Ocean). Associated principal components are significantly correlated with climate indices. The second principal component shows a correlation of about 60% with the MEI index. On the other hand, the third principal component presents a 55% correlation with the PDO index. This indicates that the steric-corrected sea level still contains important interannual variability, possibly due to the manometric mass component (not corrected here). Understanding this residual signal is an

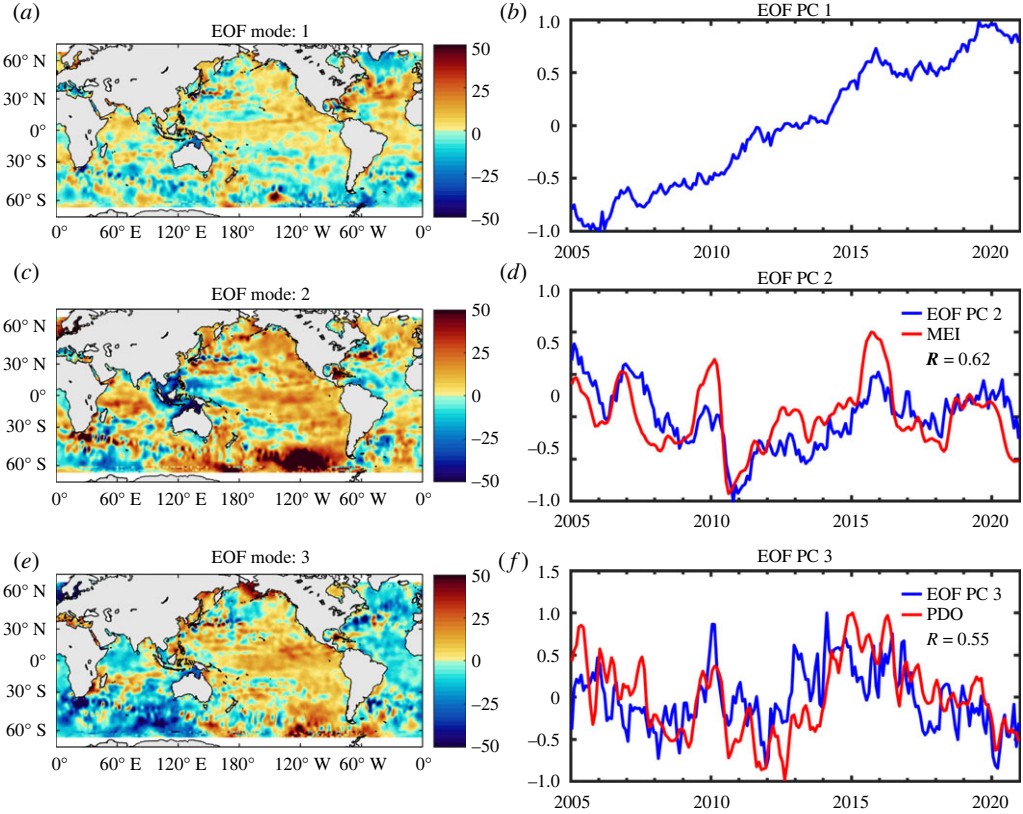

**Figure 10.** EOF decomposition of altimetry-based sea-level corrected for steric effects over 2005–2020 (same dataset from [122] as used for figure 9). Spatial patterns (*a,c,e*); principal components (PC) (*b,d,f*). Modes 1, 2 and 3 are shown, along with time series of the MEI and PDO climate indices. *R* is the correlation coefficient between PCs and climate indices. (Online version in colour.)

important goal for improved knowledge of the involved processes as well as for decadal and longer-term sea-level projections from climate model simulations.

## (d) Regional sea-level acceleration

The acceleration of the regional sea-level rise has also been investigated [97,111]. Based on a similar approach as in Ablain *et al.* [24] for the altimetry error budget on the GMSL, Prandi *et al.* [142] suggest uncertainties of $0.83\,\mathrm{mm\,yr^{-1}}$ for regional trends and $0.062\,\mathrm{mm\,yr^{-2}}$ for regional accelerations (90% confidence level).

   Most recent results (O Andersen & S Nerem 2021, personal communication) suggest that the altimetry-based regional sea-level acceleration is roughly uniform across the oceans, as expected, if mostly due to accelerated ice sheet melting in response to anthropogenic global warming (see §4).

## 4. Anthropogenic forcing versus natural variability (detection and attribution)

An intriguing question is whether the accelerating global mean sea-level evolution and the spatial trend patterns are evidence of anthropogenic forcing (from greenhouse gas (GHG) emissions and aerosols), or from natural and/or internal climate variability. This issue is often called 'Detection and Attribution'. Detection indicates a statistically significant change compared with a previous state while attribution refers to the identification of the causes for the observed change. On the

basis of comparisons between observed sea level and components, and climate simulations with and without external forcings (both natural and anthropogenic), studies have demonstrated that contemporary GMSL rise cannot be explained by internal climate variability only (e.g. [147–151]). These studies showed that since 1970, anthropogenic forcing accounts for approximately 70% of the observed sea-level rise (e.g. [150]), with an increasing contribution beyond that date. Similar approaches have been applied to individual components (e.g. ocean thermal expansion [152]; glaciers [153], with similar conclusions (see [151] for a review)).

At regional scale, the role of anthropogenic forcing has been much debated (e.g. [102,126,151,154]), although most recent studies find convincing evidence of the emergence of forced patterns in regional sea-level observations (e.g. [108,155,156]). Being still largely driven by internal climate modes, regional sea-level trends are much larger than the GMSL rise [125]. Thus, the forced anthropogenic signal is difficult to detect due to low signal-to-noise ratio (noise meaning here internal climate variability [110]). Comparing spatial trend patterns in sea level with multi ensembles of Earth System Models over the altimetry era, Fasullo & Nerem [108]) found similarity in the observed spatial patterns and those simulated with anthropogenic forcing in almost all oceanic regions, but dominantly in the Southern Ocean. Moreover, Fasullo *et al.* [156] were able to separate the respective roles of individual forcing agents (GHGs and aerosols) depending on the regions and their changing contributions over time (e.g. increasing GHGs versus decreasing aerosols). Using future climate model simulations, other studies focused on the time of emergence of the forced signal from the internal climate variability (e.g. [63,157]) and found it will be detectable in more than 50% of all oceanic areas by 2040–2050.

The approximately 30-year long altimetry-based spatial trend patterns now look quite stable, suggesting that these begin indeed to reflect the forced signal. This is of crucial importance to evaluate the impacts of sea-level rise in vulnerable coastal areas, since locally, sea-level rise is the superposition of the global mean plus the regional variability and local small-scale processes (see §5). Knowing that in some oceanic regions, regional trends now reflect the long-term forced response is important information to take into account for prevention and adaptation purposes.

## 5. Coastal sea level

In the coastal zones, sea level results from the superposition of the global mean rise, large-scale regional changes and small-scale coastal processes. Local VLMs also contribute to sea-level change relatively to the ground. Along most of the world's coastlines, sea level remains so far poorly measured and thus understood [158,159]. The main reasons are: (1) uneven distribution of the world tide gauge network, dominated by the northern hemisphere, (2) gaps in many existing tide gauge records or too short records preventing from estimating long-term coastal sea-level trends in many locations, (3) lack of valid sea-level data within approximately 15 km to the coast from classical altimetry and (4) poor knowledge of small-scale coastal processes causing local departure from regional changes.

Concerning (3), high-precision radar altimetry has been optimized for open ocean monitoring. When the satellite approaches the coast, the Earth surface illuminated by the radar (the radar footprint, of several kilometres in diameter) contains not only reflections from the ocean but also from the nearby land. This considerably perturbs the altimetry-based estimate of the sea surface height. As a consequence, the number of valid altimetry-based sea-level data drops from more than 90% in the open ocean to near zero in the 15 km-wide coastal zone [160–162]. In addition, the potential existence of small-scale coastal processes able to change sea-level trends close to the coast compared with the open ocean, in theory prevents extrapolating the altimetry-based gridded sea-level data (of typically 1/4° resolution) up to the coast. Regarding (4), a few recent studies have indeed stressed the importance of small-scale coastal phenomena on coastal sea-level trends (e.g. [8]). These include local atmospheric effects, baroclinic instabilities, coastal trapped waves, shelf currents, waves and fresh water input from rivers in estuaries.

For example, natural climate modes such as ENSO and IOD induce eastward propagating equatorial Kelvin waves affecting coastlines in the equatorial Pacific and Indian oceans [127]. In

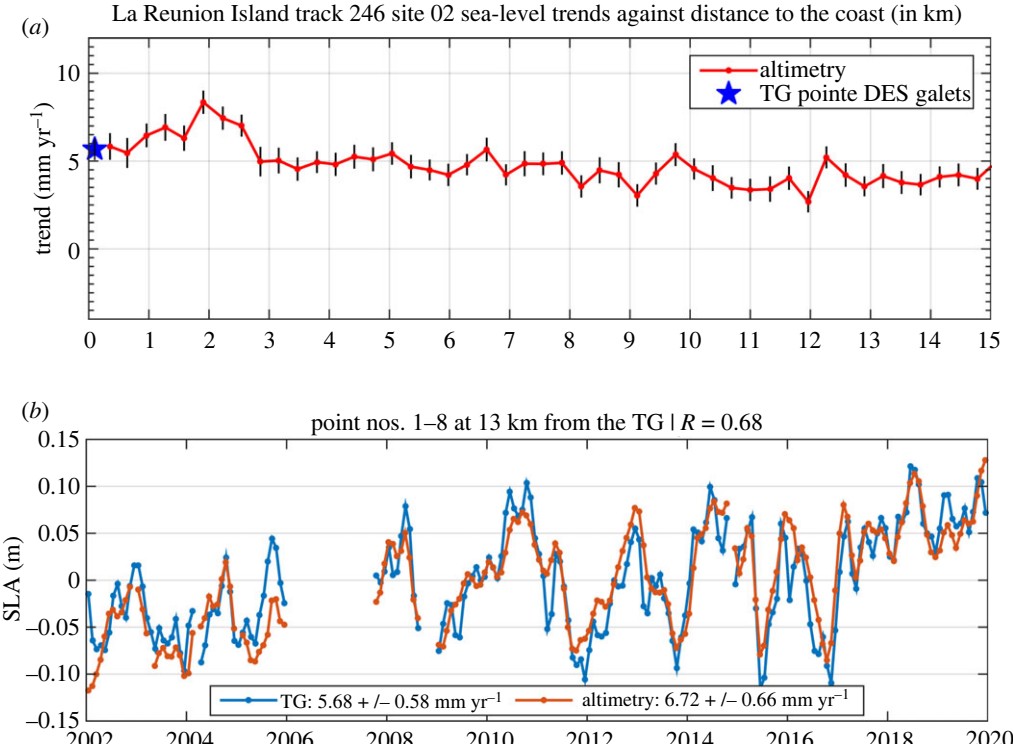

**Figure 11.** (*a*) Along-track coastal sea-level trend (red curve) from reprocessed Jason-1, 2 and 3 altimetry data over January 2002–December 2019 at La Reunion Island (Indian Ocean) against distance to the coast (in km). The black vertical bars are the 1-sigma errors. The blue star on the left represents the sea-level trend over the same time span at the tide gauge located 13 km away from the satellite track. (*b*) Comparison of the altimetry-based sea-level time series (along-track averaged over the first 2 km from the coast) and the tide gauge record over January 2002–December 2019. TG means tide gauge. R is the correlation coefficient between the altimetry and tide gauge sea-level time series. (Online version in colour.)

the Atlantic Ocean, possible links between the NAO and coastal sea level have been questioned [127]. Changes in coastal sea level may also result from changes in coastal ocean circulation driven by bathymetry, shape of coastal boundaries and changing forcing factors, e.g. trend in wind stress. Fresh water discharge to the coastal ocean delivered by rivers in deltas and estuaries is another process able to produce sea-level variations at the coast through water mass and density changes [163,164]. Wind-generated waves caused by changes in atmospheric circulation can also play a significant role in changing sea level very close to the coast [165–167]. Due to all involved processes, coastal sea level may eventually greatly vary from one location to the other. Their importance relative to offshore processes still needs to be quantified.

A first step of improvement consists of revisiting the potential of altimetry measurements in the world coastal zones to obtain valid estimates of present-day sea level in the nearshore vicinity. A recent effort in this direction has been recently carried out [168,169], consisting of reprocessing raw radar echoes of several successive altimetry missions (an approach called 'retracking' [170]) in order to extract, as accurately as possible, the satellite range measurement (height of the satellite above the sea surface) from which the coastal sea surface height is deduced. This reprocessing has allowed to retrieve valid coastal sea-level data very close to the coast (less than 6 km) at about more than 750 sites located along the coasts of North and South America, Northeast Atlantic, Mediterranean Sea, Africa, North Indian Ocean, Asia and Australia. For nearly 300 sites, the closest distance to the coast with valid sea-level data is now less than 3.5 km, sometimes less

than 1 km. An example is presented in figure 11 showing sea-level trends over January 2002–December 2019 against distance to the coast at La Reunion Island (Indian Ocean), as well as a comparison of the reprocessed sea-level time series close to the coast with the tide gauge record. This new coastal dataset (freely available from the SEANOE website, https://doi.org/10.17882/74354) provides a network of about 750 altimetry-based 'virtual' coastal stations, along the world coastlines, complementing the current network of tide gauges [169].

# 6. Future sea level

With ongoing and future global warming, sea level will continue to rise. Successive IPCC reports have provided process-based projections of future sea-level change under different global warming scenarios. The warming scenarios of previous IPCC reports (e.g. AR5 [171]) were based on Representative Concentration Pathways (RCPs; expressing the total radiative forcing by 2100 relative to pre-industrial levels, for different scenarios of GHG concentrations). In general, three RCPs are considered from low to high emissions: RCP2.6, RCP4.5 and RCP8.5. In the IPCC AR6, more realistic warming scenarios are considered, the Shared Socioeconomic Pathways (SSPs), ranked from SSP1 (sustainability) to SSP5 (fossil fuel development) in order to represent different socio-economic developments as well as different pathways of atmospheric GHG concentrations. SSP5 roughly coincides with RCP8.5.

The sea-level projections are based on individual estimates of the steric and mass components by coupled climate model simulations. Over time, projections of the steric component have not much changed. This is because processes causing future temperature and salinity changes as well as ocean circulation changes are rather well understood. For example, the IPCC 4th (AR4, [172]), 5th (AR5, [171]) [172] and 6th (AR6, [6]) assessment reports projected a steric contribution to sea level of 0.3, $0.27 \pm 0.06$ and $0.3 \pm 0.06$ m, respectively, by 2100 (with respect to the early 2000s) for the larger warming scenario. The projected glacier contribution has either not much changed (0.15, $0.16 \pm 0.07$ and $0.18 \pm 0.03$ m by 2100 for AR4, AR5 and AR6, respectively). This is unlike the ice sheet contribution, which has been revised upward in the SROCC report [4] and IPCC AR6 [6] compared with AR4 and AR5, a result of improved understanding and modelling of the ice sheet dynamics, in particular, the marine ice sheet instabilities, not accounted for in previous AR4 and AR5 reports. The projected conservative contributions of Greenland and Antarctica amount to $0.13 \pm 0.04$ and $0.12 \pm 0.09$ m, respectively, in the AR6 for the SSP5 scenario in 2100. However, projections accounting for marine ice cliff instabilities (MICI) in Antarctica (a process proposed by DeConto & Pollard [173], although as yet unobserved and questioned by Edwards *et al.* [174]) would raise the Antarctic contribution to $0.34 \pm 0.15$ m in 2100 (SSP5). Finally, the land water storage is expected to not contribute by more than $0.03 \pm 0.01$ m in 2100. Summing up all components leads to an average total contribution to the GMSL of $0.77 \pm 0.07$ and $0.99 \pm 0.08$ m in 2100 without and with MICI [6].

In recent years, a number of probabilistic projections have also been published (e.g. [175–181]). This approach assumes probability distributions for each component of the GMSL that account for increased uncertainty ranges compared with conventional physical (process-based) models. Some of these probabilistic projections lead to higher sea-level rise than process-based models. For example, Kopp *et al.* [177] suggest a GMSL rise in 2100 of $1.46 \pm 0.0.37$ m, a value significantly higher than in the IPCC AR6.

Regional projections have also been provided (e.g. [171,182,183]). Regional discrepancies with respect to the global mean rise result from ocean circulation changes and thermal expansion, as well as sea-level fingerprints due to gravitational and rotational deformations in response to future mass redistributions and GIA. The associated spatial patterns have not significantly changed since the AR5 [171]. They consist of an amplification of the GMSL in the tropics by 20–30%, a regional increase in the Northwest Atlantic (mostly due to GIA and decrease of the AMOC) and in the Austral Ocean.

Longer-term projections have also been recently proposed, although with very high uncertainty ranges because of obvious large uncertainties in ice sheet dynamics and warming

scenarios. The IPCC AR6 [6] report a GMSL rise (without MICI) of 2.2 to 5.9 m in 2300 with respect to the early 2000s for the SSP5 scenario.

## 7. Concluding remarks and perspectives

Owing to spectacular improvements of various global observing systems from space and *in situ*, the sea-level science has incredibly progressed during the last three decades. Precise monitoring of sea-level changes and components has been made available, which in turn has allowed improved understanding of the processes at work. In parallel, important progress in modelling has been achieved, leading to rather good agreement between observations and model products, and improved confidence in projections for future changes. However, several issues still need progress, in particular considering that sea-level rise is one of the most threatening consequences of future global warming. We summarize below what would be needed for improved sea-level science in the near future in terms of observations and modelling at global, regional and local scales (e.g. [32,184,185] and [158] for more details).

### (a) GMSL and budget closure

To assess the global mean sea-level budget, we need sustained observations with as global as possible coverage of sea level by satellite altimetry and of the steric and mass components from various space-based and *in situ* observing systems. This includes continuity of the high-precision altimetry record beyond the Sentinel-6 Michael Frielich mission launched in November 2020. It would also be important to maintain the level of quality of the historical past missions (through regular data reprocessing) to ensure the homogeneity of the time series. Monitoring of sea level in the Arctic is also of high priority. Continuity of the ESA CryoSat-2 polar mission (https://www.esa.int/Applications/Observing_the_Earth/FutureEO/CryoSat) until the launch of new high-latitude missions is crucial. For the steric component, priorities include continuing support for Core Argo and operational implementation of Deep Argo, subsurface temperature measurements (e.g. using dedicated Argo floats or other autonomous devices, and ship measurements) in areas not well covered (e.g. marginal seas, high latitudes, boundary currents, and shallow areas and shelf regions). For the mass component, sustained measurements of ocean mass changes, of ice sheet and glaciers mass balances, and of land water storage changes from GRACE-type missions with improved performances, especially in terms of spatial resolution, are highly needed. In addition, sustained monitoring of land ice bodies using other observing systems (InSAR, radar and optical imagery, standard radar, SAR and laser altimetry), as well as modelling, are equally important. Concerning the terrestrial water contribution, continuing improvement of global hydrological models including human activities (ground water extraction, dam building on rivers, land use change, etc.) remains an important challenge. Assimilation of space-based and *in situ* data into the models (e.g. altimetry-based stream flow estimates, GRACE-based mass changes, etc.) may greatly contribute to hydrological model improvement.

### (b) Regional sea level

Understanding regional sea-level changes is not as advanced as for the global mean sea-level rise. This mostly comes from two factors: (1) remaining important uncertainties in gridded altimetry-based sea level and steric data and (2) broader variety of processes causing regional sea-level changes and still limited understanding of their drivers (e.g. for regional steric changes). Progress may come from various directions: (1) characterization of observational uncertainties of the various components of the regional budget (altimetry-based trends, Argo-based steric trends, GRACE-based ocean mass trends, atmospheric pressure trends) and plans to reduce them below the $1 \, \mathrm{mm \, yr^{-1}}$ level regionally; (2) assessment of closure/non-closure of the regional sea-level budget and identification of areas where the budget is not closed with thorough investigations of potential causes; (3) estimate of the dominant contributions (i.e. thermosteric, halosteric, ocean

mass, atmospheric loading) depending on areas (northern, tropical, southern and coastal oceans) and analysis of the dominant forcing factors (e.g. wind stress versus heat and fresh water fluxes); (4) revisit of detection and attribution issues at regional scale in order to evaluate the possibility of emergence of the anthropogenic forcing depending on regions as well as unambiguous identification of the fingerprint signal.

## (c) Coastal sea level

The world's coastal zones are under increasing stress due to natural phenomena and human activities. These include: extreme events, shoreline retreat, loss of biodiversity, pollution, sea-level rise and ground subsidence due to water and oil and gas extraction. Coastal zones are the most densely populated areas in the world, with 11 of the 15 largest megacities located near the sea. Coastal zones are areas of important human activities (e.g. fishing, industries, tourism) and in many regions, coastal ecosystems provide not only food but a variety of services. While up to now, climate-related sea-level rise has remained modest (a few tens of cm at most since the beginning of the twentieth century), it may become a major threat (together with extreme events) in the coming decades, considering that more than 600 million people are living at less than 10 m above sea level, a number that may double before the end of the twenty-first century [10]. Precisely measuring present-day sea-level rise at as many as possible coastal sites and simulating accurate projections for the future decades/centuries are major goals for the scientific community, considering the urgent demand from stakeholders in charge of coastal planning, adaptation and mitigation. In parallel to observational progress, development of very high-resolution coastal ocean models (currently lacking in most of the world coastal oceans) is crucial for understanding small-scale coastal processes occurring along coastlines and interpreting coastal sea-level observations.

Finally, since what matters at the coast in terms of societal impact is relative sea-level rise (e.g. [9]), measuring vertical land motions at the coast using GNSS and InSAR tools is as important as measuring absolute sea-level changes, since at many low-lying coastlines, ground subsidence due to ground water or gas extraction as well as sediment loading and compaction can greatly amplify climate-related sea-level rise and the associated societal impacts [10,13].

Data accessibility. This article has no additional data.

Authors' contributions. A.C.: conceptualization, supervision, writing—original draft, writing—review and editing; L.M.: data curation, software and validation.

Both authors gave final approval for publication and agreed to be held accountable for the work performed therein.

Conflict of interest declaration. We declare we have no competing interests.

Funding. L.M. is supported by a post-doctoral fellowship from the International Space Science Institute (ISSI, Bern Switzerland).

Acknowledgements. A.C. thanks The Royal Society for encouraging her to write this review. A.C. and L.M. thank the Editor, Chris Garrett, as well as three reviewers, Marta Marcos, Carolina Camargo and Jerry Mitrovica for useful comments that helped to improve the original version of the manuscript. They also thank Matthew Palmer and Carolina Camargo for providing figures from their published papers, as well as Michael Ablain for interesting discussions about errors of altimetry systems and Robin Fraudeau for preparing figure 3. A.C. and L.M. are also very grateful to the many authors of the papers quoted in the reference list for their considerable work performed along the years on sea-level science.

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
