## [Peer Review File · Proceedings. Mathematical, Physical, and Engineering Sciences]

Review History

RSPA-2022-0049.R0 (Original submission)

Review form: Referee 1

Is the manuscript an original and important contribution to its field?

Excellent

Is the paper of sufficient general interest?

Excellent

Is the overall quality of the paper suitable?

Excellent

Can the paper be shortened without overall detriment to the main message?

Yes

Do you think some of the material would be more appropriate as an electronic appendix?

No

Do you have any ethical concerns with this paper?

No

Recommendation?

Accept with minor revision (please list in comments)

Comments to the Author(s)

This review manuscript provides a comprehensive overview of sea level observations, variability and processes, including the most updated literature. It addresses the main topics of current research in sea-level science, from global to regional, from open ocean to the coast, and focusing on the altimetry period. The structure of the paper is coherent and clear, and the figures are illustrative of the different sections. I have really enjoyed reading this manuscript; even if I am mostly up to date with the current literature on sea level, I have discovered new recent papers that are of interest. I believe this manuscript is useful for sea-level scientists, but also for researchers working on ocean or climate science who are not necessarily that familiar with the topic and with the large number of recent publications. I think that it certainly deserves publication.

I am listing below minor corrections (mostly typos) and suggestions for the authors:

- Page 1, line 43: delete 2nd "occurring" (appears twice and is not necessary)
- P. 1, l. 50: "sea level variations". Please, consider hyphenate sea-level in this context and throughout the paper
- P. 1, l. 59: remove "also"
- P. 2, l. 35: can now be precisely
- P. 3, l. 50: in which -> to which?
- p.6, l. 34: launch -> launched
- p. 7, l. 10: This is why...
- p. 7, l. 24: occurred->occurred
- p. 8, fig. 4: shouldn't this figure be moved to the section on regional sea level?
- P. 10, l. 45: 1913->1993
- P. 12, l. 50-51: it would be good to provide a number on the possible contribution of Thwaites, for comparison to others above.
- P. 13, l. 33: because it is completely...
- P. 16, l. 19: Dangendorf et al (2019) reports acceleration since late 1960s
- P. 18, l. 43: play -> plays
- P. 21, section 3.1.2: I think it would be good to provide an estimate of its magnitude (i.e., the IB approach)
- P. 21, l. 38: except in formerly glaciated high-latitude regions
- P. 22, l. 57: separate principal components
- P. 25, l. 14: result->results
- P. 26, l. 7: maybe add Piecuch et al 2018 <https://www.pnas.org/content/115/30/7729>
- P. 26, l. 8: "induced by anthropogenic climate change": actually, waves contribute irrespective of the origin of changes in winds
- P. 26, l. 14: need -> needs
- Section 5: little is said about the importance of VLM at the coast. Perhaps is worth to extend the issue in this section.
- P. 27, l. 48: patthaways->pathways
- P. 28, l. 55: report->reports
- P. 30, on regional sea level: another research direction could be to improve observations in shallow waters and mechanisms of exchange between open ocean and marginal and shallow seas
- P. 31: one major need for coastal sea level studies is knowledge of coastal bathymetries as these are of major relevance for coastal processes

Review form: Referee 2

Is the manuscript an original and important contribution to its field?

Good

Is the paper of sufficient general interest?

Excellent

Is the overall quality of the paper suitable?

Good

Can the paper be shortened without overall detriment to the main message?

Yes

Do you think some of the material would be more appropriate as an electronic appendix?

No

Do you have any ethical concerns with this paper?

No

Recommendation?

Accept with minor revision (please list in comments)

Comments to the Author(s)

Review of "Contemporary Sea level Changes from Global to Local Scales: A Review"

The manuscript is a review of recent advances of sea-level science during the satellite altimetry era (since 1993). Given the amount of works published in the recent years, the review is a useful publication and tool for readers wanting to become familiar with the topic. Thus, I believe this review deserves to be published.

I do have some general comments and more specific comments (attached pdf of the manuscript) that I would like the authors to address:

General comments:

- I think some extra works can be included in the review. I've mentioned them through the text where I think it's suitable.

- Some terms deserve to be better explained, as a review is usually used by people 'entering' the field. For example, the authors mention steric sea-level, and then talk about ocean temperature and salinity datasets, but haven't drawn the link between these two things before. Another example is the appearance of terms such as 'geoid height' and 'dynamic height' in Figure 2 that are not explained in the text. These are not trivial informations, and should be explained to the reader.

- For most of the regional section, I was getting the impression that the authors wanted to convey the message that the GRD fingerprints don't contribute much to regional SLC. Only at the end of the section 3.1.4. it became clear that the authors believe that the altimetry-steric residual contain important contributions of the GRD fingerprints. I think some sentences could be rewritten to make it clear the message that they are important contributors to regional SLC, because if someone doesn't see those two last sentences of section 3.1.4, the wrong message can be conveyed.

- Some terms need to be better explained to avoid confusion. For example, 'ocean mass' is used both to refer to GMSL change due to mass loss of glaciers, ice sheets and TWS, and for regional SLC due to ocean circulation. If the author use 'barystatic' SLC for the first case, and 'dynamic SL' (as part of sterodynamic component) it will become clearer.

- Section 3.1.2 can be expanded.

Figures:

- Avoid jet colormaps: they are not colorblind friendly, and can lead to wrong conclusions.
- Figure 4 lower panel is a bit confusing and misleading. I suggest the authors to clarify that they are showing the different in relation to the GMSL trend, as one can quickly get wrong conclusions if they just scroll through the paper.
- Figure 9 could benefit from a third panel, showing the difference between altimetry and steric maps.

Editorial comments:

- Avoid short paragraphs, with only 1 or 2 sentences.
- I think its satellite altimeter, and not 'altimeter satellite'

Minor comments:

Addressed in the attached pdf.

Review form: Referee 3

Is the manuscript an original and important contribution to its field?

Excellent

Is the paper of sufficient general interest?

Excellent

Is the overall quality of the paper suitable?

Excellent

Can the paper be shortened without overall detriment to the main message?

Yes

Do you think some of the material would be more appropriate as an electronic appendix?

No

Do you have any ethical concerns with this paper?

No

Recommendation?

Accept with minor revision (please list in comments)

Comments to the Author(s)

See attached review

Decision letter (RSPA-2022-0049.R0)

22-Mar-2022

Dear Dr Cazenave,

On behalf of the Reviews Editor, I am pleased to inform you that your Manuscript RSPA-2022-0049 entitled "Contemporary Sea level Changes from Global to Local Scales :A Review" has been accepted for publication subject to minor revisions in Proceedings A. Please find the referees' comments below.

The reviewer(s) have recommended publication, but also suggest some minor revisions to your manuscript. Therefore, I invite you to respond to the reviewer(s)' comments and revise your manuscript. We normally ask for minor revisions to be submitted within 7 days but if you think you will need longer than this then please let me know.

To revise your manuscript, log into <https://mc.manuscriptcentral.com/prsa> and enter your Author Centre, where you will find your manuscript title listed under "Manuscripts with Decisions." Under "Actions," click on "Create a Revision." Your manuscript number has been appended to denote a revision.

You will be unable to make your revisions on the originally submitted version of the manuscript. Instead, revise your manuscript and upload a new version through your Author Centre.

When submitting your revised manuscript, you will be able to respond to the comments made by the referee(s) and upload a file "Response to Referees" in Step 1: "View and Respond to Decision Letter". Please provide a point-by-point response to the comments raised by the reviewers and the editor(s). A thorough response to these points will help us to assess your revision quickly. You can also upload a 'tracked changes' version either as part of the 'Response to reviews' or as a 'Main document'.

IMPORTANT: Your original files are available to you when you upload your revised manuscript. Please delete any redundant files before completing the submission process.

When uploading your revised files, please make sure that you include the following as we cannot proceed without these:

1) A text file of the manuscript (doc, txt, rtf or tex), including the references, tables (including captions) and figure captions. Please remove any tracked changes from the text before submission. PDF files are not an accepted format for the "Main Document".

2) A separate electronic file of each figure (tif, eps or print-quality pdf preferred). The format should be produced directly from original creation package, or original software format.

3) Electronic Supplementary Material (ESM): all supplementary materials accompanying an accepted article will be treated as in their final form. Note that the Royal Society will not edit or typeset supplementary material and it will be hosted as provided. Please ensure that the supplementary material includes the paper details where possible (authors, article title, journal name). Supplementary files will be published alongside the paper on the journal website and posted on the online figshare repository (<https://figshare.com>). The heading and legend provided for each supplementary file during the submission process will be used to create the figshare page, so please ensure these are accurate and informative so that your files can be found in searches. Files on figshare will be made available approximately one week before the accompanying article so that the supplementary material can be attributed a unique DOI.

Alternatively you may upload a zip folder containing all source files for your manuscript as described above with a PDF as your "Main Document". This should be the full paper as it appears when compiled from the individual files supplied in the zip folder.

Article Funder

Please ensure you fill in the Article Funder question on page 2 to ensure the correct data is collected for FundRef (<http://www.crossref.org/fundref/>).

Media summary

Please ensure you include a short non-technical summary (up to 100 words) of the key findings/importance of your paper. This will be used for to promote your work and marketing purposes (e.g. press releases). The summary should be prepared using the following guidelines:

*Write simple English: this is intended for the general public. Please explain any essential technical terms in a short and simple manner.

*Describe (a) the study (b) its key findings and (c) its implications.

*State why this work is newsworthy, be concise and do not overstate (true 'breakthroughs' are a rarity).

*Ensure that you include valid contact details for the lead author (institutional address, email address, telephone number).

Cover images

We welcome submissions of images for possible use on the cover of Proceedings A. Images should be square in dimension and please ensure that you obtain all relevant copyright permissions before submitting the image to us. If you would like to submit an image for consideration please send your image to proceedingsa@royalsociety.org

Open Access

As a Perspective article this will be made open access free of charge.

Once again, thank you for submitting your manuscript to Proceedings A and I look forward to receiving your revision. If you have any questions at all, please do not hesitate to get in touch.

Best wishes
Raminder Shergill
proceedingsa@royalsociety.org
Proceedings A

on behalf of
Dr Chris Garrett
Reviews Editor
Proceedings A

Reviewer(s)' Comments to Author:

Referee: 1 (Marta Marcos)

Comments to the Author(s)

This review manuscript provides a comprehensive overview of sea level observations, variability and processes, including the most updated literature. It addresses the main topics of current research in sea-level science, from global to regional, from open ocean to the coast, and focusing

on the altimetry period. The structure of the paper is coherent and clear, and the figures are illustrative of the different sections. I have really enjoyed reading this manuscript; even if I am mostly up to date with the current literature on sea level, I have discovered new recent papers that are of interest. I believe this manuscript is useful for sea-level scientists, but also for researchers working on ocean or climate science who are not necessarily that familiar with the topic and with the large number of recent publications. I think that it certainly deserves publication.

I am listing below minor corrections (mostly typos) and suggestions for the authors:

- Page 1, line 43: delete 2nd “occurring” (appears twice and is not necessary)
- P. 1, l. 50: “sea level variations”. Please, consider hyphenate sea-level in this context and throughout the paper
- P. 1, l. 59: remove “also”
- P. 2, l. 35: can now be precisely
- P. 3, l. 50: in which -> to which?
- p.6, l. 34: launch -> launched
- p. 7, l. 10: This is why...
- p. 7, l. 24: occurred->occurred
- p. 8, fig. 4: shouldn't this figure be moved to the section on regional sea level?
- P. 10, l. 45: 1913->1993
- P. 12, l. 50-51: it would be good to provide a number on the possible contribution of Thwaites, for comparison to others above.
- P. 13, l. 33: because it is completely...
- P. 16, l. 19: Dangendorf et al (2019) reports acceleration since late 1960s
- P. 18, l. 43: play -> plays
- P. 21, section 3.1.2: I think it would be good to provide an estimate of its magnitude (i.e., the IB approach)
- P. 21, l. 38: except in formerly glaciated high-latitude regions
- P. 22, l. 57: separate principal components
- P. 25, l. 14: result->results
- P. 26, l. 7: maybe add Piecuch et al 2018 <https://www.pnas.org/content/115/30/7729>
- P. 26, l. 8: “induced by anthropogenic climate change”: actually, waves contribute irrespective of the origin of changes in winds
- P. 26, l. 14: need -> needs
- Section 5: little is said about the importance of VLM at the coast. Perhaps is worth to extend the issue in this section.
- P. 27, l. 48: patthaways->pathways
- P. 28, l. 55: report->reports
- P. 30, on regional sea level: another research direction could be to improve observations in shallow waters and mechanisms of exchange between open ocean and marginal and shallow seas
- P. 31: one major need for coastal sea level studies is knowledge of coastal bathymetries as these are of major relevance for coastal processes

Referee: 2

Comments to the Author(s)

Review of “Contemporary Sea level Changes from Global to Local Scales: A Review”

The manuscript is a review of recent advances of sea-level science during the satellite altimetry era (since 1993). Given the amount of works published in the recent years, the review is a useful publication and tool for readers wanting to become familiar with the topic. Thus, I believe this review deserves to be published.

I do have some general comments and more specific comments (attached pdf of the manuscript) that I would like the authors to address:

General comments:

- I think some extra works can be included in the review. I've mentioned them through the text where I think it's suitable.
- Some terms deserve to be better explained, as a review is usually used by people 'entering' the field. For example, the authors mention steric sea-level, and then talk about ocean temperature and salinity datasets, but haven't drawn the link between these two things before. Another example is the appearance of terms such as 'geoid height' and 'dynamic height' in Figure 2 that are not explained in the text. These are not trivial informations, and should be explained to the reader.
- For most of the regional section, I was getting the impression that the authors wanted to convey the message that the GRD fingerprints don't contribute much to regional SLC. Only at the end of the section 3.1.4. it became clear that the authors believe that the altimetry-steric residual contain important contributions of the GRD fingerprints. I think some sentences could be rewritten to make it clear the message that they are important contributors to regional SLC, because if someone doesn't see those two last sentences of section 3.1.4, the wrong message can be conveyed.
- Some terms need to be better explained to avoid confusion. For example, 'ocean mass' is used both to refer to GMSL change due to mass loss of glaciers, ice sheets and TWS, and for regional SLC due to ocean circulation. If the author use 'barystatic' SLC for the first case, and 'dynamic SL' (as part of sterodynamic component) it will become clearer.
- Section 3.1.2 can be expanded.

Figures:

- Avoid jet colormaps: they are not colorblind friendly, and can lead to wrong conclusions.
- Figure 4 lower panel is a bit confusing and misleading. I suggest the authors to clarify that they are showing the different in relation to the GMSL trend, as one can quickly get wrong conclusions if they just scroll through the paper.
- Figure 9 could benefit from a third panel, showing the difference between altimetry and steric maps.

Editorial comments:

- Avoid short paragraphs, with only 1 or 2 sentences.
- I think its satellite altimeter, and not 'altimeter satellite'

Minor comments:

Addressed in the attached pdf.

Referee: 3 (Jerry Mitrovica)

Comments to the Author(s)

See attached review

Reviews Editor Comments:

Dear Anny,

I apologise for the long time it has taken to obtain referee reports on your nice review paper, but I hope that you will find the wait worthwhile, in that we now have three reports, all of which I think you will find helpful.

All three referees recommend acceptance with minor revision. I therefore invite you to make changes at your discretion. Please prepare responses to the points raised in the reports, for transmission to the referees along with the revised paper though we will not call for further review.

Among many other comments in a report and on an annotated copy of your paper, referee 2 calls for more explanation of the meaning of 'steric' changes. You first introduce the term in Section 2.2.1 without defining it. Later in the paper you mention thermosteric and halosteric components, which I find confusing. This may be a naïve question, but how can there be a halosteric component, at least globally, without the addition, or loss, of freshwater, which counts as a mass change? Is there a danger of double-dipping? A quick Google search for 'halosteric' leads to a few publications, though I haven't read them.

As a further example of my ignorance, which may be shared by some readers, I didn't find the word 'eustatic' anywhere in your paper, though this is a term that many newcomers to the field might expect to see defined. Also, I think readers might welcome your views on the possibility of 'black swan' events. (I don't know whether the metaphor carries over into French as 'cygne noir'!) You do make a passing reference to the Thwaites Glacier, but perhaps more could be said about this and other possibilities.

You will note that two of the referees reveal their identities. I expect that you plan to thank them, and the anonymous referee, in the acknowledgements. There's certainly no need to mention me. Many congratulations again on this, and many thanks to you and Lorena Moreira for preparing an excellent review paper.

Best wishes,
Chris Garrett

Author's Response to Decision Letter for (RSPA-2022-0049.R0)

See Appendix A.

Decision letter (RSPA-2022-0049.R1)

19-Apr-2022

Dear Dr Cazenave

On behalf of the Reviews Editor, I am pleased to inform you that your manuscript entitled "Contemporary Sea level Changes from Global to Local Scales :A Review" has been accepted in its final form for publication in Proceedings A.

Our Production Office will be in contact with you in due course. You can expect to receive a proof of your article soon. Please contact the office to let us know if you are likely to be away from e-mail in the near future. If you do not notify us and comments are not received within 5 days of sending the proof, we may publish the paper as it stands.

As a reminder, you have provided the following 'Data accessibility statement' (if applicable). Please remember to make any data sets live prior to publication, and update any links as needed when you receive a proof to check. It is good practice to also add data sets to your reference list.
Statement (if applicable):

Under the terms of our licence to publish you may post the author generated postprint (ie. your accepted version not the final typeset version) of your manuscript at any time and this can be made freely available. Postprints can be deposited on a personal or institutional website, or a recognised server/repository. Please note however, that the reporting of postprints is subject to a media embargo, and that the status the manuscript should be made clear. Upon publication of the definitive version on the publisher's site, full details and a link should be added.

You can cite the article in advance of publication using its DOI. The DOI will take the form: 10.1098/rspa.XXXX.YYYY, where XXXX and YYYY are the last 8 digits of your manuscript number (eg. if your manuscript number is RSPA-2017-1234 the DOI would be 10.1098/rspa.2017.1234).

For tips on promoting your accepted paper see our blog post:
<https://royalsociety.org/blog/2020/07/promoting-your-latest-paper-and-tracking-your-results/>

Thank you for your submission. On behalf of the Editors of the journal, we look forward to your continued contributions to the Journal.

Best wishes
Raminder Shergill,
Proceedings A Editorial Office
proceedingsa@royalsociety.org

on behalf of
Dr Chris Garrett
Reviews Editor
Proceedings A

Responses to Reviewers and Editor ‘ Comments*(in italics bold)***Referee: 1 (Marta Marcos)**

Comments to the Author(s)

This review manuscript provides a comprehensive overview of sea level observations, variability and processes, including the most updated literature. It addresses the main topics of current research in sea-level science, from global to regional, from open ocean to the coast, and focusing on the altimetry period. The structure of the paper is coherent and clear, and the figures are illustrative of the different sections. I have really enjoyed reading this manuscript; even if I am mostly up to date with the current literature on sea level, I have discovered new recent papers that are of interest. I believe this manuscript is useful for sea-level scientists, but also for researchers working on ocean or climate science who are not necessarily that familiar with the topic and with the large number of recent publications. I think that it certainly deserves publication.

I am listing below minor corrections (mostly typos) and suggestions for the authors:

- Page 1, line 43: delete 2nd “occurring” (appears twice and is not necessary)
- P. 1, l. 50: “sea level variations”. Please, consider hyphenate sea-level in this context and throughout the paper
- P. 1, l. 59: remove “also”
- P. 2, l. 35: can now be precisely
- P. 3, l. 50: in which -> to which?
- p.6, l. 34: launch -> launched
- p. 7, l. 10: This is why...
- p. 7, l. 24: occurred->occurred
- p. 8, fig. 4: shouldn't this figure be moved to the section on regional sea level?
- P. 10, l. 45: 1913->1993
- P. 12, l. 50-51: it would be good to provide a number on the possible contribution of Thwaites, for comparison to others above.
- P. 13, l. 33: because it is completely...
- P. 16, l. 19: Dangendorf et al (2019) reports acceleration since late 1960s
- P. 18, l. 43: play -> plays
- P. 21, section 3.1.2: I think it would be good to provide an estimate of its magnitude (i.e., the IB approach)
- P. 21, l. 38: except in formerly glaciated high-latitude regions
- P. 22, l. 57: separate principal components
- P. 25, l. 14: result->results
- P. 26, l. 7: maybe add Piecuch et al 2018 <https://www.pnas.org/content/115/30/7729>
- P. 26, l. 8: “induced by anthropogenic climate change”: actually, waves contribute irrespective of the origin of changes in winds
- P. 26, l. 14: need -> needs
- Section 5: little is said about the importance of VLM at the coast. Perhaps is worth to extend the issue in this section.
- P. 27, l. 48: patthaways->pathways
- P. 28, l. 55: report->reports
- P. 30, on regional sea level: another research direction could be to improve observations in shallow waters and mechanisms of exchange between open ocean and marginal and shallow seas

- P. 31: one major need for coastal sea level studies is knowledge of coastal bathymetries as these are of major relevance for coastal processes

We thank Marta Marcos for her comments. We have taken into account all minor comments.

Referee: 2 (Carolina Camargo)

Comments to the Author(s)

Review of “Contemporary Sea level Changes from Global to Local Scales: A Review”

The manuscript is a review of recent advances of sea-level science during the satellite altimetry era (since 1993). Given the amount of works published in the recent years, the review is a useful publication and tool for readers wanting to become familiar with the topic. Thus, I believe this review deserves to be published.

I do have some general comments and more specific comments (attached pdf of the manuscript) that I would like the authors to address:

We thank Reviewer 2 for all her comments.

General comments:

- I think some extra works can be included in the review. I’ve mentioned them through the text where I think it’s suitable.

- Some terms deserve to be better explained, as a review is usually used by people ‘entering’ the field. For example, the authors mention steric sea-level, and then talk about ocean temperature and salinity datasets, but haven’t drawn the link between these two things before. Another example is the appearance of terms such as ‘geoid height’ and ‘dynamic height’ in Figure 2 that are not explained in the text. These are not trivial informations, and should be explained to the reader.

We have modified the text to explain what is the steric sea level (as well as the thermosteric and halosteric components). The quantities appearing in Fig.2 have been defined.

- For most of the regional section, I was getting the impression that the authors wanted to convey the message that the GRD fingerprints don’t contribute much to regional SLC. Only at the end of the section 3.1.4. it became clear that the authors believe that the altimetry-steric residual contain important contributions of the GRD fingerprints. I think some sentences could be rewritten to make it clear the message that they are important contributors to regional SLC, because if someone doesn’t see those two last sentences of section 3.1.4, the wrong message can be conveyed.

We have added a sentence to mention that in the future the GRD fingerprints will be important. But it remains that at present they are almost undetectable (except may be around Greenland).

- Some terms need to be better explained to avoid confusion. For example, ‘ocean mass’ is used both to refer to GMSL change due to mass loss of glaciers, ice sheets and TWS, and for

regional SLC due to ocean circulation. If the author use ‘barystatic’ SLC for the first case, and ‘dynamic SL’ (as part of sterodynamic component) it will become clearer.

This has been clarified and relevant terms have been defined

- Section 3.1.2 can be expanded.

Section 3.1.2 (now 3.2.3) has been expanded.

Figures:

- Avoid jet colormaps: they are not colorblind friendly, and can lead to wrong conclusions.
- Figure 4 lower panel is a bit confusing and misleading. I suggest the authors to clarify that they are showing the different in relation to the GMSL trend, as one can quickly get wrong conclusions if they just scroll through the paper.

We now clearly explain that the bottom map of Fig.4 (now Fig.8b) represents trends with respect to the global mean trend.

- Figure 9 could benefit from a third panel, showing the difference between altimetry and steric maps.

This has been added

Editorial comments:

- Avoid short paragraphs, with only 1 or 2 sentences.
- I think its satellite altimeter, and not ‘altimeter satellite’

Corrected

Minor comments:

Addressed in the attached pdf.

We accounted for all comments made by Rev.2 on the manuscript

Referee: 3 (Jerry Mitrovica)

Review of “Contemporary Sea Level Changes from Global to Local Scales” by Anny Cazenave and Lorena Moreira. This manuscript provides a broad overview of the major issues in modern sea level research, with a focus on the spectrum of spatial and temporal variability in observations and the underlying physical processes, both natural and anthropogenic. As the authors note, there have been other recent reviews of the topic, including one I am involved in (Hamlington et al., 2020b) - the present review is somewhat less technical than the others, but I think that will make it more interesting to a much wider scientifically-literate audience. The authors are clearly up to the task – the senior author is of course a central figure in the field and the junior author is responsible for some superb recent work that I am becoming familiar with. I have a few comments on the text, which I list below.

We thank Jerry Mitrovica for his comments

- Page 4 discusses recent estimates of the rate of 20th century GMSL change. The text ends with the sentences: “The recent study by Palmer et al. (2021) considers five recent reconstructions in an ensemble approach to quantify the 20th century mean sea level rise and its uncertainty. They estimate the mean sea level elevation of 12 +/- 5 cm between 1901 and 1990 with a mean rate of rise of 1.3 +/- 0.6 mm/yr over the period (Fig.1). This new estimate is significantly lower than some previous ones (e.g., 1.9 mm/yr, Jevrejeva et al., 2014) but substantially larger than some others (e.g., 1.1 mm/yr, Hay et al., 2015).” I think this sentence overstates the current uncertainty (and underestimates the current consensus) in this estimate. First of all, the Hay et al. (2015) estimate is consistent with the independent statistical analysis of Dangendorf et al. (2017) which yielded a value of 1.1 +/- 0.3 mm/yr. Both analyses deal rather effectively with VLM effects, so the “dispersion” in estimates that the authors mention is a likely issue with the Jevrejeva et al. (2014) estimate, which is widely seen as too high – an outlier. I might add that the text above makes the strange argument that 1.3 +/- 0.6 mm/yr is “substantially larger” than 1.1 mm/yr!

We have modified the text to make this clearer.

- Early in the paper the authors discuss the issue that altimetry is measuring a different sea level change than tide gauges but they then quote GMSL values for both – and a non-expert reader will not differentiate between the two. It is important that when Figure 3 is introduced that the reader is aware that the label on the figure is actually a different quantity than the label on Figure 2. In this regard, the authors should reference two papers: Frederikse et al. (GRL, 2017) and Lickley et al. (J. Clim., 2018) in the discussion of the difference. Both papers point out the bias that may be introduced when assessing global ocean mass changes using altimetry records – and that bias can be even larger if one is considering regional rates.

When talking about altimetry, we use the term ‘global mean sea level’ while when talking about tide gauges we refer to ‘mean sea level’. In addition for the historical reconstruction we mention that they are corrected for VLMs, thus comparable to what altimetry provides.

- A few points about section 3.1.3. (i) The response of the solid Earth to ongoing ice melt doesn’t just depend on “lithosphere elasticity” – it depends on elasticity of the entire mantle, not just the lithosphere; (ii) I would delete “(e.g., in the tropics)” because the rotational effect on sea level skews the fingerprint field such that some locations near the tropics don’t necessarily see a rise; (iii) I am not sure I would agree with “so far mostly known from modeling” – perhaps “predicted by modeling”; (iv) the GIA mention of the sea level equation misses key citations – Farrell and Clark (GJI, 1976) especially, but also Mitrovica and Milne (GJI, 2003), and I would add Lambeck et al. (PNAS, 2014); Tamisiea (2011) should be cited in the -0.3 mm/yr value.

We have modified the text to account for this comment and have added the suggested references.

- I reread the Moreira et al. (2021b) paper by the same authors but had forgotten that it argues that 25% of the signal “in the vicinity of the Greenland ice sheet” can be explained by the fingerprint signal (p. 22). To be more precise, the authors might revise this phrasing to read

“in the south-eastern Greenland region” (to quote from the 2021b paper). Our group has a paper submitted on a related topic (Dr. Cazenave was suggested as a reviewer!) – I will have the first author send on that submitted paper.

Thanks

Other points:

- I think it would be useful to provide a few definitions to aid the reader – some examples are phrases like “dynamical coastal instabilities” (p. 12)

Done

- I don’t think GRACE provides a ”direct” measure of the mass of ice loss (p. 11,12) since it also sees solid Earth deformation;

Like all observing systems GRACE needs to be corrected for unrelated processes (e.g., GIA). When corrected, GRACE is the most direct measurement of land ice mass (although with a poor resolution...).

- The caption to Fig. 9 should describe the data used to create the bottom panel;

We have not added the references to the different steric data sets because this would have made the figure caption too long but refer to Moreira et al. 2021b for information.

- The references to artificial damming of water might include the reference Hawley et al. (Earth’s Future, 2020).

Added

Editor Comments:

All three referees recommend acceptance with minor revision. I therefore invite you to make changes at your discretion. Please prepare responses to the points raised in the reports, for transmission to the referees along with the revised paper though we will not call for further review.

Among many other comments in a report and on an annotated copy of your paper, referee 2 calls for more explanation of the meaning of ‘steric’ changes. You first introduce the term in Section 2.2.1 without defining it. Later in the paper you mention thermosteric and halosteric components, which I find confusing. This may be a naïve question, but how can there be a halosteric component, at least globally, without the addition, or loss, of freshwater, which counts as a mass change? Is there a danger of double-dipping? A quick Google search for ‘halosteric’ leads to a few publications, though I haven’t read them.

We added details on the definition of the steric sea level and its two components : the thermosteric component (due to temperature change only) and haolosteric component due

to salinity changes only). We quote the paper by J Gregory (Gregory et al., 20129) who define the steric sea level (this can also be found in many other articles of the literature).

We also add a sentence saying that fresh addition to the ocean only changes the global mean ocean mass (but not global mean salt content). In Gregory et al. (2019), Appendix 2 explains why :

Extrat from Gregory et al. (2019) :

« Appendix 2: Why We Can Ignore Global Halosteric Sea-Level Change ?

When freshwater enters the ocean, such as from melting continental ice sheets, it adds to the ocean mass and in turn increases global-mean sea level (barystatic sea-level rise). Ocean salinity also changes due to the dilution of sea water, thus suggesting a role for a global halosteric sea-level change (Munk 2003; Levitus et al. 2005). However, the net effect on global-mean sea level is almost entirely barystatic since the global halosteric effect is negligible (Lowe and Gregory 2006). We can understand why this is so by recognizing that freshwater entering the ocean sees its salinity increase while the ambient sea water is itself freshened. These compensating salinity changes (which are often ignored, as by Munk 2003 and Levitus et al. 2005) have corresponding compensating sea-level changes, thus bringing the global halosteric effect to near zero. We demonstrate this effect in the following subsections, by considering a two-bucket thought experiment where one bucket holds freshwater (bucket-1) and the other holds sea water (bucket-2). We ask how the total water volume changes upon homogenizing the water in the two buckets, while conserving the masses of freshwater and salt. As we will see, the total volume of homogenized water is very nearly equal to the sum of the volume initially in the two separate buckets (to within 0:1%). »

As a further example of my ignorance, which may be shared by some readers, I didn't find the word 'eustatic' anywhere in your paper, though this is a term that many newcomers to the field might expect to see defined.

For many years now, it is recommended to no more use the term 'eustatic' in sea level sciences because it has been misused in the past (I attach the paper by J. Gregory who discuss that issue (bottom of page 1270 and page 1278)).

Also, I think readers might welcome your views on the possibility of 'black swan' events. (I don't know whether the metaphor carries over into French as 'cygne noir'!) You do make a passing reference to the Thwaites Glacier, but perhaps more could be said about this and other possibilities.

The paragraph on the Thwaites Glacier had disappeared from the first revised version. This was a mistake. It is now reintroduced and expanded (with now 4 references).